# Accelerating Motion Planning via Optimal Transport

**An T. Le**[1], **Georgia Chalvatzaki**[1,3,5], **Armin Biess, Jan Peters**[1−4]
[1]Department of Computer Science, Technische Universitat Darmstadt, Germany
[2]German Research Center for AI (DFKI)    [3]Hessian.AI    [4]Centre for Cognitive Science
[5]Center for Mind, Brain and Behavior, Uni. Marburg and JLU Giessen, Germany

## Abstract

Motion planning is still an open problem for many disciplines, e.g., robotics, autonomous driving, due to their need for high computational resources that hinder real-time, efficient decision-making. A class of methods striving to provide smooth solutions is gradient-based trajectory optimization. However, those methods usually suffer from bad local minima, while for many settings, they may be inapplicable due to the absence of easy-to-access gradients of the optimization objectives. In response to these issues, we introduce Motion Planning via Optimal Transport (MPOT)—a *gradient-free* method that optimizes a batch of smooth trajectories over highly nonlinear costs, even for high-dimensional tasks, while imposing smoothness through a Gaussian Process dynamics prior via the planning-as-inference perspective. To facilitate batch trajectory optimization, we introduce an original zero-order and highly-parallelizable update rule—-the Sinkhorn Step, which uses the regular polytope family for its search directions. Each regular polytope, centered on trajectory waypoints, serves as a local cost-probing neighborhood, acting as a *trust region* where the Sinkhorn Step "transports" local waypoints toward low-cost regions. We theoretically show that Sinkhorn Step guides the optimizing parameters toward local minima regions of non-convex objective functions. We then show the efficiency of MPOT in a range of problems from low-dimensional point-mass navigation to high-dimensional whole-body robot motion planning, evincing its superiority compared to popular motion planners, paving the way for new applications of optimal transport in motion planning.

## 1  Introduction

Motion planning is a fundamental problem for various domains, spanning robotics [1], autonomous driving [2], space-satellite swarm [3], protein docking [4]. etc., aiming to find feasible, smooth, and collision-free paths from start-to-goal configurations. Motion planning has been studied both as sampling-based search [5, 6] and as an optimization problem [7–9]. Both approaches have to deal with the complexity of high-dimensional configuration spaces, e.g., when considering high-degrees of freedom (DoF) robots, the multi-modality of objectives due to multiple constraints at both configuration and task space, and the requirement for smooth trajectories that low-level controllers can effectively execute. Sampling-based methods sample the high-dimensional manifold of configurations and use different search techniques to find a feasible and optimal path [6, 10, 5], but suffer from the complex sampling process and the need for large computational budgets to provide a solution, which increases w.r.t. the complexity of the problem (e.g., highly redundant robots and narrow passages) [11]. Optimization-based approaches work on a trajectory level, optimizing initial trajectory samples either via covariant gradient descent [7, 12] or through probabilistic inference [9, 8, 13]. Nevertheless, as with every optimization pipeline, trajectory optimization depends on initialization and can get trapped in bad local minima due to the non-convexity of complex objectives. Moreover, in some problem settings, objective gradients are unavailable or expensive to compute. Indeed, trajectory optimization is difficult to tune and is often avoided in favor of sampling-based methods with probabilistic completeness. We refer to Appendix H for an extensive discussion of related works.

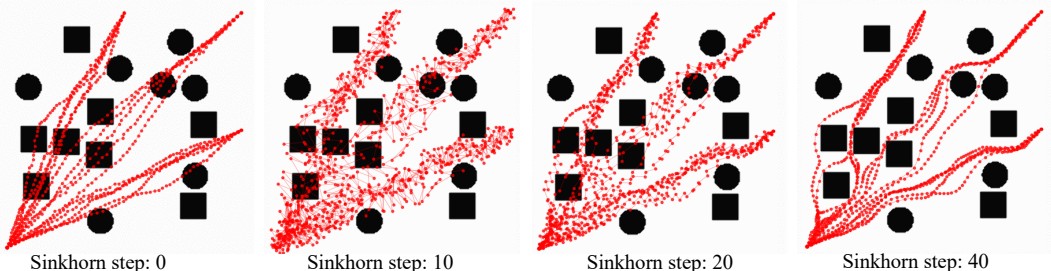

| Sinkhorn step: 0 | Sinkhorn step: 10 | Sinkhorn step: 20 | Sinkhorn step: 40 |

Figure 1: Example of MPOT in the multimodal planar navigation scenario with three different goals. For each goal, we sample five initial trajectories from a GP prior. We illustrate four snapshots of our proposed Sinkhorn Step that updates a batch of waypoints from multiple trajectories over multiple goals. For this example, the **total planning time was 0.12s**. More demos can be found on https://sites.google.com/view/sinkhorn-step/

To address these issues of trajectory optimization, we propose a zero-order, fast, and highly parallelizable update rule—the Sinkhorn Step. We apply this novel update rule in trajectory optimization, resulting in *Motion Planning via Optimal Transport (MPOT)* – a gradient-free trajectory optimization method optimizing a batch of smooth trajectories. MPOT optimizes trajectories by solving a sequence of entropic-regularized Optimal Transport (OT) problems, where each OT instance is solved efficiently with the celebrated Sinkhorn-Knopp algorithm [14]. In particular, MPOT discretizes the trajectories into waypoints and structurally probes a local neighborhood around each of them, which effectively exhibits a *trust region*, where it "transports" local waypoints towards low-cost areas given the local cost approximated by the probing mechanism. Our method is simple and does not require computing gradients from cost functions propagating over long kinematics chains. Crucially, the planning-as-inference perspective [15, 13] allows us to impose constraints related to transition dynamics as planning costs, additionally imposing smoothness through a Gaussian Process (GP) prior. Delegating complex constraints to the planning objective allows us to locally resolve trajectory update as an OT problem at each iteration, updating the trajectory waypoints towards the local optima, thus effectively optimizing for complex cost functions formulated in configuration and task space. We also provide a preliminary theoretical analysis of the Sinkhorn Step, highlighting its core properties that allow optimizing trajectories toward local minima regions.

Further, our empirical evaluations on representative tasks with high-dimensionality and multimodal planning objectives demonstrate an increased benefit of MPOT, both in terms of planning time and success rate, compared to notable trajectory optimization methods. Moreover, we empirically demonstrate the convergence of MPOT in a 7-DoF robotic manipulation setting, showcasing a fast convergence of MPOT, reflected also in its dramatically reduced planning time w.r.t. baselines. The latter holds even for 36-dimensional, highly redundant mobile manipulation systems in long-horizon fetch and place tasks (cf. Fig. 4).

Our **contribution** is twofold. *(i)* We propose the Sinkhorn Step - an efficient zero-order update rule for optimizing a batch of parameters, formulated as a barycentric projection of the current points to the polytope vertices. *(ii)* We, then, apply the Sinkhorn Step to motion planning, resulting in a novel trajectory optimization method that optimizes a batch of trajectories by efficiently solving a sequence of linear programs. It treats every waypoint across trajectories equally, enabling fast batch updates of multiple trajectories-waypoints over multiple goals by solving a single OT instance while retaining smoothness due to integrating the GP prior as cost function.

## 2 Preliminaries

**Entropic-regularized optimal transport.** We briefly introduce discrete OT. For a thorough introduction, we refer to [16–18].

**Notation.** Throughout the paper, we consider the optimization on a $d$-dimensional Euclidean space $\mathbb{R}^d$, representing the parameter space (e.g., a system state space). $\mathbf{1}_d$ is the vector of ones in $\mathbb{R}^d$. The scalar product for vectors and matrices is $x, y \in \mathbb{R}^d$, $\langle x, y \rangle = \sum_{i=1}^d x_i y_i$; and $\boldsymbol{A}, \boldsymbol{B} \in \mathbb{R}^{d \times d}$, $\langle \boldsymbol{A}, \boldsymbol{B} \rangle = \sum_{i,j=1}^d \boldsymbol{A}_{ij} \boldsymbol{B}_{ij}$, respectively. $\|\cdot\|$ is the $l_2$-norm, and $\|\cdot\|_{\boldsymbol{M}}$ denotes the Mahalanobis norm w.r.t. some positive definite matrix $\boldsymbol{M} \succ 0$. For two histograms $\boldsymbol{n} \in \Sigma_n$ and $\boldsymbol{m} \in \Sigma_m$ in the simplex $\Sigma_d := \{\boldsymbol{x} \in \mathbb{R}_+^d : \boldsymbol{x}^\mathsf{T} \mathbf{1}_d = 1\}$, we define the set $U(\boldsymbol{n}, \boldsymbol{m}) := \{\boldsymbol{W} \in \mathbb{R}_+^{n \times m} \mid \boldsymbol{W} \mathbf{1}_m = \boldsymbol{n}, \boldsymbol{W}^\mathsf{T} \mathbf{1}_n = \boldsymbol{m}\}$ containing $n \times m$ matrices with row

and column sums $\boldsymbol{n}$ and $\boldsymbol{m}$ respectively. Correspondingly, the entropy for $\boldsymbol{A} \in U(\boldsymbol{n}, \boldsymbol{m})$ is defined as $H(\boldsymbol{A}) = -\sum_{i,j=1}^{n,m} a_{ij} \log a_{ij}$.

Let $\boldsymbol{C} \in \mathbb{R}_+^{n \times m}$ be the positive cost matrix, the OT between $\boldsymbol{n}$ and $\boldsymbol{m}$ given cost $\boldsymbol{C}$ is $\text{OT}(\boldsymbol{n}, \boldsymbol{m}) := \min_{\boldsymbol{W} \in U(\boldsymbol{n}, \boldsymbol{m})} \langle \boldsymbol{W}, \boldsymbol{C} \rangle$. Traditionally, OT does not scale well with high dimensions. To address this, Cuturi [19] proposes to regularize its objective with an entropy term, resulting in the entropic-regularized OT

$$\text{OT}_\lambda(\boldsymbol{n}, \boldsymbol{m}) := \min_{\boldsymbol{W} \in U(\boldsymbol{n}, \boldsymbol{m})} \langle \boldsymbol{W}, \boldsymbol{C} \rangle - \lambda H(\boldsymbol{W}). \tag{1}$$

Solving (1) with Sinkhorn-Knopp [19] has a complexity of $\tilde{O}(n^2/\epsilon^3)$ [20], where $\epsilon$ is the approximation error w.r.t. the original OT. Higher $\lambda$ enables a faster but "blurry" solution, and vice versa.

**Trajectory optimization.** Given a parameterized trajectory by a discrete set of support states and control inputs $\boldsymbol{\tau} = [\mathbf{x}_0, \boldsymbol{u}_0, ..., \mathbf{x}_{T-1}, \boldsymbol{u}_{T-1}, \mathbf{x}_T]^\intercal$, trajectory optimization aims to find the optimal trajectory $\boldsymbol{\tau}^*$, which minimizes an objective function $c(\boldsymbol{\tau})$, with $\mathbf{x}_0$ being the start state. Standard motion planning costs, such as goal cost $c_g$ defined as the distance to a desired goal-state $\boldsymbol{x}_g$, obstacle avoidance cost $c_{obs}$, and smoothness cost $c_{sm}$ can be included in the objective. Hence, trajectory optimization can be expressed as the sum of those costs while obeying the dynamics constraint

$$\boldsymbol{\tau}^* = \arg\min_{\boldsymbol{\tau}} [c_{obs}(\boldsymbol{\tau}) + c_g(\boldsymbol{\tau}, \mathbf{x}_g) + c_{sm}(\boldsymbol{\tau})] \text{ s.t. } \dot{\boldsymbol{x}} = f(\mathbf{x}, \boldsymbol{u}) \text{ and } \boldsymbol{\tau}(0) = \boldsymbol{x}_0. \tag{2}$$

For many manipulation tasks with high-DoF robots, this optimization problem is typically highly non-linear due to many complex objectives and constraints. Besides $c_{obs}$, $c_{sm}$ is crucial for finding smooth trajectories for better tracking control. Covariant Hamiltonian Optimization for Motion Planning (CHOMP) [7] designs a finite difference matrix $\boldsymbol{M}$ resulting to the smoothness cost $c_{sm} = \boldsymbol{\tau}^\intercal \boldsymbol{M} \boldsymbol{\tau}$. This smoothness cost can be interpreted as a penalty on trajectory derivative magnitudes. Mukadam et al. [8] generalizes the smoothness cost by incorporating a GP prior as cost via the planning-as-inference perspective [15, 21], additionally constraining the trajectories to be dynamically smooth. Recently, an emergent paradigm of multimodal trajectory optimization [22, 23, 13, 24] is promising for discovering different modes for non-convex objectives, thereby exhibiting robustness against bad local minima. Our work contributes to this momentum by proposing an efficient batch update-rule for vectorizing waypoint updates across timesteps and number of plans.

## 3 Sinkhorn Step

To address the problem of batch trajectory optimization in a gradient-free setting, we propose *Sinkhorn Step*—a zero-order update rule for a batch of optimization variables. Our method draws inspiration from the free-support barycenter problem [25], where the mean support of a set of empirical measures is optimized w.r.t. the OT cost. Consider an optimization problem with some objective function without easy access to function derivatives. This barycenter problem can be utilized as a parameter update mechanism, i.e., by defining a set of discrete target points (i.e., local search directions) and a batch of optimizing points as two empirical measures, the barycenter of these empirical measures acts as the updated optimizing points based on the objective function evaluation at the target points.

With these considerations in mind, we introduce *Sinkhorn Step*, consisting of two components: a polytope structure defining the unbiased search-direction bases, and a weighting distribution for evaluating the search directions. Particularly, the weighting distribution has row-column unit constraints and must be efficient to compute. Following the motivation of [25], the entropic-regularized OT fits nicely into the second component, providing a solution for the weighting distribution as an OT plan, which is solved extremely fast, and its solution is unique [19]. In this section, we formally define Sinkhorn Step and perform a preliminary theoretical analysis to shed light on its connection to *directional-direct search* methods [26, 27], thereby motivating its algorithmic choices and practical implementation proposed in this paper.

### 3.1 Problem formulation

We consider the batch optimization problem

$$\min_X f(X) = \min_X \sum_{i=1}^n f(\boldsymbol{x}_i), \tag{3}$$

where $X = \{\boldsymbol{x}_i\}_{i=1}^n$ is a set of $n$ optimizing points, $f : \mathbb{R}^d \to \mathbb{R}$ is non-convex, differentiable, bounded below, and has $L$-Lipschitz gradients.

**Assumption 1.** *The objective $f$ is $L$-smooth with $L > 0$*

$$\|\nabla f(\boldsymbol{x}) - \nabla f(\boldsymbol{y})\| \leq L \|\boldsymbol{x} - \boldsymbol{y}\|, \forall \boldsymbol{x}, \boldsymbol{y} \in \mathbb{R}^d$$

*and bounded below by $f(\boldsymbol{x}) \geq f_* \in \mathbb{R}, \forall \boldsymbol{x} \in \mathbb{R}^d$.*

Throughout the paper, we assume that function evaluation is implemented batch-wise and is cheap to compute. Function derivatives are either expensive or impossible to compute. At the first iteration, we sample a set of initial points $X_0 \sim \mathcal{D}_0$, with its matrix form $\boldsymbol{X}_0 \in \mathbb{R}^{n \times d}$, from some prior distribution $\mathcal{D}_0$. The goal is to compute a batch update for the optimizing points, minimizing the objective function. This problem setting suits trajectory optimization described in Section 4.

### 3.2 Sinkhorn Step formulation

Similar to directional-direct search, Sinkhorn Step typically evaluates the objective function over a search-direction-set $D$, ensuring descent with a sufficiently small stepsize. The search-direction-set is typically a vector-set requiring to be a *positive spanning set* [28], i.e., its conic hull is $\mathbb{R}^d = \{\sum_i w_i \boldsymbol{d}_i, \boldsymbol{d}_i \in D, w_i \geq 0\}$, ensuring that every point (including the extrema) in $\mathbb{R}^d$ is reachable by a sequence of positive steps from any initial point.

**Regular Polytope Search-Directions**. Consider a $(d-1)$-unit hypersphere $S^{d-1} = \{\boldsymbol{x} \in \mathbb{R}^d : \|\boldsymbol{x}\| = 1\}$ with the center at zero.

**Definition 1** (Regular Polytope Search-Directions)**.** *Let us denote the regular polytope family $\mathcal{P} = \{simplex, orthoplex, hypercube\}$. Consider a $d$-dimensional polytope $P \in \mathcal{P}$ with $m$ vertices, the search-direction set $D^P = \{\boldsymbol{d}_i \mid \|\boldsymbol{d}_i\| = 1\}_{i=1}^m$ is constructed from the vertex set of the regular polytope $P$ inscribing $S^{d-1}$.*

The $d$-dimensional regular polytope family $\mathcal{P}$ has all of its dihedral angles equal and, hence, is an unbiased sparse approximation of the circumscribed $(d-1)$-sphere, i.e., $\sum_i \boldsymbol{d}_i = 0, \|\boldsymbol{d}_i\| = 1 \forall i$. There also exist other regular polytope families. However, the regular polytope types in $\mathcal{P}$ exist in every dimension (cf. [29]). Moreover, the algorithmic construction of general polytope is not trivial [30]. Vertex enumeration for $\mathcal{P}$ is straightforward for vectorization and simple to implement, which we found to work well in our settings–see also Appendix F. We state the connection between regular polytopes and the positive spanning set in the following proposition.

**Proposition 1.** $\forall P \in \mathcal{P}, D^P$ *forms a positive spanning set.*

This property ensures that any point $\boldsymbol{x} \in \mathbb{R}^d$, $\boldsymbol{x} = \sum_i w_i \boldsymbol{d}_i, w_i \geq 0, \boldsymbol{d}_i \in D^P$ can be represented by a positively weighted sum of the set of directions defined by the polytopes.

**Batch Update Rule**. At an iteration $k$, given the current optimizing points $X_k$ and their matrix form $\boldsymbol{X}_k \in \mathbb{R}^{n \times d}$, we first construct the direction set from a chosen polytope $P$, and denote the direction set $\boldsymbol{D}^P \in \mathbb{R}^{m \times d}$ in matrix form. Similar to [25], let us define the prior histograms reflecting the importance of optimizing points $\boldsymbol{n} \in \Sigma_n$ and the search directions $\boldsymbol{m} \in \Sigma_m$, then the constraint space $U(\boldsymbol{n}, \boldsymbol{m})$ of OT is defined. With these settings, we define Sinkhorn Step.

**Definition 2** (Sinkhorn Step)**.** *The batch update rule is the barycentric projection (Remark 4.11, [17]) that optimizes the free-support barycenter of the optimizing points and the batch polytope vertices*

$$\boldsymbol{X}_{k+1} = \boldsymbol{X}_k + \boldsymbol{S}_k, \; \boldsymbol{S}_k = \alpha_k diag(\boldsymbol{n})^{-1} \boldsymbol{W}_\lambda^* \boldsymbol{D}^P$$
$$s.t. \; \boldsymbol{W}_\lambda^* = \operatorname{argmin}_{\boldsymbol{W} \in U(\boldsymbol{n}, \boldsymbol{m})} \langle \boldsymbol{W}, \boldsymbol{C} \rangle - \lambda H(\boldsymbol{W}), \tag{4}$$

*with $\alpha_k > 0$ as the stepsize, $\boldsymbol{C} \in \mathbb{R}^{n \times m}, \boldsymbol{C}_{i,j} = f(\boldsymbol{x}_i + \alpha_k \boldsymbol{d}_j), \boldsymbol{x}_i \in X_k, \boldsymbol{d}_j \in D^P$ is the local objective matrix evaluated at the linear-translated polytope vertices.*

Observe that the matrix $\operatorname{diag}(\boldsymbol{n})^{-1} \boldsymbol{W}_\lambda^*$ has $n$ row vectors in the simplex $\Sigma_m$. The batch update *transports* $\boldsymbol{X}$ to a barycenter shaping by the polytopes with weights defined by the optimal solution $\boldsymbol{W}_\lambda^*$. However, in contrast with the barycenter problem [25], the target measure supports are constructed locally at each optimizing point, and, thus, the points are transported in accordance with their local search directions. By Proposition 1, $D^P$ is a positive spanning set, thus, $\boldsymbol{W}_\lambda^*$ forms a *generalized barycentric coordinate*, defined w.r.t. the finite set of polytope vertices. This property implies any point in $\mathbb{R}^d$ can be reached by a sequence of Sinkhorn Steps. For the $d$-simplex case, any point inside the convex hull can be identified with a unique barycentric coordinate [31], which is not the case for $d$-orthoplex or $d$-cube. However, coordinate uniqueness is not required for our analysis in this paper, given the following assumption.

**Assumption 2.** *At any iteration $k > 0$, the prior histogram on the optimizing points and the search-direction set is uniform $\boldsymbol{n} = \boldsymbol{m} = \boldsymbol{1}_n/n$, having the same dimension $n = m$. Additionally, the entropic scaling approaches zero $\lambda \to 0$.*

Assuming uniform prior importance of the optimizing points and their search directions is natural since, in many cases, priors for stepping are unavailable. However, our formulation also suggests a conditional Sinkhorn Step, which is interesting to study in future work. This assumption allows performing an analysis on Sinkhorn Step on the original OT solution.

With these assumptions, we can state the following theorem for each $\boldsymbol{x}_k \in X_k$ separately, given that they follow the Sinkhorn Step rule.

**Theorem 1** (Main result). *If Assumption 1 and Assumption 2 hold and the stepsize is sufficiently small $\alpha_k = \alpha$ with $0 < \alpha < 2\mu_P\epsilon/L$, then with a sufficient number of iterations*

$$K \geq k(\epsilon) := \frac{f(\boldsymbol{x}_0) - f_*}{(\mu_P\epsilon - \frac{L\alpha}{2})\alpha} - 1, \tag{5}$$

*we have $\min_{0 \leq k \leq K} \|\nabla f(\boldsymbol{x}_k)\| \leq \epsilon, \forall \boldsymbol{x}_k \in X_k$.*

Note that we do not make any additional assumptions on $f$ besides the smoothness and boundedness, and the analysis is performed on non-convex settings. Theorem 1 only guarantees that the gradients of some points in the sequence of Sinkhorn Steps are arbitrarily small, i.e., in the proximity of local minima. If in practice, we implement the sufficient decreasing condition $f(\boldsymbol{x}_k) - f(\boldsymbol{x}_{k+1}) \geq c\alpha_k^2$, then $f(\boldsymbol{x}_K) \leq f(\boldsymbol{x}_i), \|\nabla f(\boldsymbol{x}_i)\| \leq \epsilon$ holds. However, this sufficient decrease check may waste some iterations and worsen the performance. We show in the experiments that the algorithm empirically exhibits convergence behavior without this condition checking. If $L$ is known, then we can compute the optimal stepsize $\alpha = \mu_P\epsilon/L$, leading to the complexity bound $k(\epsilon) = \frac{2L(f(\boldsymbol{x}_0) - f_*)}{\mu_P^2\epsilon^2} - 1$.

Therefore, the complexity bounds for $d$-simplex, $d$-orthoplex and $d$-cube are $O(d^2/\epsilon^2)$, $O(d/\epsilon^2)$, and $O(1/\epsilon^2)$, respectively. The $d$-cube case shows the same complexity bound $O(1/\epsilon^2)$ as the well-known gradient descent complexity bound on the L-smooth function [32]. These results are detailed in Appendix A. Generally, we perform a preliminary study on Sinkhorn Step with Assumption 1 and Assumption 2 to connect the well-known directional-direct search literature [26, 27], as many unexplored theoretical properties of Sinkhorn Step remain in practical settings described in Section 4.

## 4 Motion Planning via Optimal Transport

Here, we introduce MPOT - a method that applies *Sinkhorn Step* to solve the batch trajectory optimization problem, where we realize waypoints in a set of trajectories as optimizing points. Due to Sinkhorn Step's properties, MPOT does not require gradients propagated from cost functions over long kinematics chains. It optimizes trajectories by solving a sequence of strictly convex linear programs with a maximum entropy objective (cf. Definition 2), smoothly transporting the waypoints according to the local polytope structure. To promote smoothness and dynamically feasible trajectories, we incorporate the GP prior as a cost via the planning-as-inference perspective.

### 4.1 Planning As Inference With Empirical Waypoint Distribution

Let us consider general discrete-time dynamics $\boldsymbol{X} = F(\mathbf{x}_0, \boldsymbol{U})$, where $\boldsymbol{X} = [\mathbf{x}_0, \ldots, \mathbf{x}_T]$ denotes the states sequence, $\boldsymbol{U} = [\boldsymbol{u}_0, ..., \boldsymbol{u}_T]$ is the control sequence, and $\mathbf{x}_0$ is the start state. The target distribution over control trajectories $\boldsymbol{U}$ can be defined as the posterior [33]

$$q(\boldsymbol{U}) = \frac{1}{Z} \exp\left(-\eta E(\boldsymbol{U})\right) q_0(\boldsymbol{U}), \tag{6}$$

with $E(\boldsymbol{U})$ the energy function representing control cost, $q_0(\boldsymbol{U}) = \mathcal{N}(\boldsymbol{0}, \boldsymbol{\Sigma})$ a zero-mean normal prior, $\eta$ a scaling term (temperature), and $Z$ the normalizing scalar.

Assuming a first-order trajectory optimization[1], the control sequence can be defined as a time-derivative of the states $\boldsymbol{U} = [\dot{\mathbf{x}}_0, ..., \dot{\mathbf{x}}_T]$. The *target posterior distribution* over both state-trajectories and their derivatives $\boldsymbol{\tau} = (\boldsymbol{X}, \boldsymbol{U}) = \{\boldsymbol{x}_t \in \mathbb{R}^d : \boldsymbol{x}_t = [\mathbf{x}_t, \dot{\mathbf{x}}_t]\}_{t=0}^T$ is defined as

$$q^*(\boldsymbol{\tau}) = \frac{1}{Z} \exp\left(-\eta c(\boldsymbol{\tau})\right) q_F(\boldsymbol{\tau}), \tag{7}$$

---

[1]We describe first-order formulation for simplicity. However, this work can be extended to second-order systems similar to [8].

which is similar to Eq. (6) with the energy function $E = c \circ F(\mathbf{x}_0, \boldsymbol{U})$ being the composition of the cost $c$ over $\boldsymbol{\tau}$ and the dynamics $F$. The dynamics $F$ is also now integrated into the prior distribution $q_F(\boldsymbol{\tau})$. The absorption of the dynamics into the prior becomes evident when we represent the prior as a zero-mean constant-velocity GP prior $q_F(\boldsymbol{\tau}) = \mathcal{N}(\mathbf{0}, \boldsymbol{K})$, with a constant time-discretization $\Delta t$ and the time-correlated trajectory covariance $\boldsymbol{K}$, as described in Appendix B.

Now, to apply Sinkhorn Step, consider the trajectory we want to optimize $\boldsymbol{\tau} = \{\boldsymbol{x}_t\}_{t=1}^{T}$, we can define the *proposal trajectory distribution* as a waypoint empirical distribution

$$p(\boldsymbol{x}; \boldsymbol{\tau}) = \sum_{t=1}^{T} p(t)p(\boldsymbol{x}|t) = \sum_{t=1}^{T} n_t \delta_{\boldsymbol{x}_t}(\boldsymbol{x}), \tag{8}$$

with the histogram $\boldsymbol{n} = [n_1, \ldots, n_T]$, $p(t) = n_t = 1/T$, and $\delta_{\boldsymbol{x}_t}$ the Dirac on waypoints at time steps $t$. In this case, we consider the model-free setting for the proposal distribution. Indeed, this form of proposal trajectory distribution typically assumes no temporal or spatial (i.e., kinematics) correlation between waypoints. This assumption is also seen in [7, 9] and can be applied in a wide range of robotics applications where the system model is fully identified. We leverage this property for batch-wise computations and batch updates over all waypoints. The integration of model constraints in the proposal distribution is indeed interesting but is deferred for future work.

Following the planning-as-inference perspective, the motion planning problem can be formulated as the minimization of a Kullback–Leibler (KL) divergence between the proposal trajectory distribution $p(\boldsymbol{x}; \boldsymbol{\tau})$ and the target posterior distribution Eq. (7) (i.e., the I-projection)

$$\begin{aligned}
\boldsymbol{\tau}^* &= \underset{\boldsymbol{\tau}}{\arg\min} \left\{ \mathrm{KL}\left(p(\boldsymbol{x}; \boldsymbol{\tau}) \,\|\, q^*(\boldsymbol{\tau})\right) = \mathbb{E}_p\left[\log q^*(\boldsymbol{\tau})\right] - H(p) \right\} \\
&= \underset{\boldsymbol{\tau}}{\arg\min} \, \mathbb{E}_p\left[ \eta c(\boldsymbol{\tau}) + \frac{1}{2}\|\boldsymbol{\tau}\|_{\boldsymbol{K}}^2 - \log Z \right] \\
&= \underset{\boldsymbol{\tau}}{\arg\min} \sum_{t=0}^{T-1} \eta \underbrace{c(\boldsymbol{x}_t)}_{\text{state cost}} + \frac{1}{2}\underbrace{\|\boldsymbol{\Phi}_{t,t+1}\boldsymbol{x}_t - \boldsymbol{x}_{t+1}\|_{\boldsymbol{Q}_{t,t+1}^{-1}}^2}_{\text{transition model cost}},
\end{aligned} \tag{9}$$

with $\boldsymbol{\Phi}_{t,t+1}$ the state transition matrix, and $\boldsymbol{Q}_{t,t+1}$ the covariance between time steps $t$ and $t+1$ originated from the GP prior (cf. Appendix B), and the normalizing constant of the target posterior $Z$ is absorbed. Note that the entropy of the empirical distribution is constant $H(p) = -\int_{\boldsymbol{x}\in\mathbb{R}^d} \frac{1}{T}\sum_{t=1}^{T} \delta_{\boldsymbol{x}_t}(\boldsymbol{x})\log p(\boldsymbol{x}; \boldsymbol{\tau}) = \log T$. Evidently, KL objective Eq. (9) becomes a standard motion planning problem Eq. (2) with the defined waypoint empirical distributions. Note that this objective is not equivalent to Eq. (3) due to the second coupling term. However, we demonstrate in Section 5.2 that MPOT still exhibits convergence. Indeed, investigating Sinkhorn Step in a general graphical model objective [34] is vital for future work. We apply Sinkhorn Step to Eq. (9) by realizing the trajectory as a batch of optimizing points $\boldsymbol{\tau} \in \mathbb{R}^{T\times d}$. This realization also extends naturally to a batch of trajectories described in the next section.

The main goal of this formulation is to naturally inject the GP dynamics prior to MPOT, benefiting from the GP sparse Markovian structure resulting in the second term of the objective Eq. (9). This problem formulation differs from the moment-projection objective [33, 8, 13], which relies on importance sampling from the proposal distribution to perform parameter updates. Contrarily, we do not encode the model in the proposal distribution and directly optimize for the trajectory parameters, enforcing the model constraints in the cost.

### 4.2 Practical considerations for applying Sinkhorn Step

For the practical implementation, we make the following realizations to the Sinkhorn Step implementation for optimizing a trajectory $\boldsymbol{\tau}$. First, we define a set of probe points for denser function evaluations (i.e., cost-to-go for each vertex direction). We populate equidistantly *probe* points along the directions in $D^P$ outwards till reaching a *probe radius* $\beta_k \geq \alpha_k$, resulting in the *probe set* $H^P$ with its matrix form $\boldsymbol{H}^P \in \mathbb{R}^{m\times h\times d}$ with $h$ probe points for each direction (cf. Fig. 2). Second, we add stochasticity in the search directions by applying a random $d$-dimensional rotation $\boldsymbol{R} \in SO(d)$ to the polytopes to promote local exploration (computation of $\boldsymbol{R} \in SO(d)$ is discussed in Appendix G). Third, to further decouple the correlations between the waypoints updates, we sample the rotation matrices in batch and then construct the direction sets from the rotated polytopes, resulting in the

tensor $\boldsymbol{D}^P \in \mathbb{R}^{T \times m \times d}$. Consequently, the *probe set* is also constructed in batch for every waypoint $\boldsymbol{H}^P \in \mathbb{R}^{T \times m \times h \times d}$. The Sinkhorn Step is computed with the *einsum* operation along the second dimension (i.e., the $m$-dimension) of $\boldsymbol{D}^P$ and $\boldsymbol{H}^P$. In intuition, the second and third considerations facilitate random permutation of the rows of the OT cost matrix.

With these considerations, the element of the $t^{\text{th}}$-waypoint and $i^{\text{th}}$-search directions in the OT cost matrix $\boldsymbol{C} \in \mathbb{R}^{T \times m}$ is the mean of probe point evaluation along a search direction (i.e., cost-to-go)

$$\boldsymbol{C}_{t,i} = \frac{1}{h} \sum_{j=1}^{h} \eta \, c(\boldsymbol{x}_t + \boldsymbol{y}_{t,i,j}) + \tfrac{1}{2} \| \boldsymbol{\Phi}_{t,t+1} \boldsymbol{x}_t - (\boldsymbol{x}_{t+1} + \boldsymbol{y}_{t+1,i,j}) \|_{\boldsymbol{Q}_{t,t+1}^{-1}}^2 , \qquad (10)$$

with the probe point $\boldsymbol{y}_{t,i,j} \in H^P$. Then, we ensure the cost matrix positiveness for numerical stability by subtracting its minimum value. With uniform prior histograms $\boldsymbol{n} = \mathbf{1}_T/T$, $\boldsymbol{m} = \mathbf{1}_m/m$, the problem $\boldsymbol{W}^* = \operatorname{argmin} \operatorname{OT}_\lambda(\boldsymbol{n}, \boldsymbol{m})$ is instantiated and solved with the log-domain stabilization version [35, 36] of the Sinkhorn algorithm. By setting a moderately small $\lambda = 0.01$ to balance between performance and blurring bias, the update does not always collapse towards the vertices of the polytope, but to a conservative one inside the polytope convex hull. In fact, the Sinkhorn Step defines an *explicit trust region*, which bounds the update inside the polytope convex hull. More discussions of log-domain stabilization and trust region properties are in Appendix E and Appendix D.

In the trajectory optimization experiments, we typically do not assume any cost structure (e.g., non-smooth, non-convex). In MPOT, Assumption 2 is usually violated with $T \gg m$, but MPOT still works well due to the soft assignment of Sinkhorn distances. We observe that finer function evaluations, randomly rotated polytopes, and moderately small $\lambda$ increase the algorithm's robustness against practical conditions. Note that these implementation technicalities do not incur much overhead due to the efficient batch computation of modern GPU.

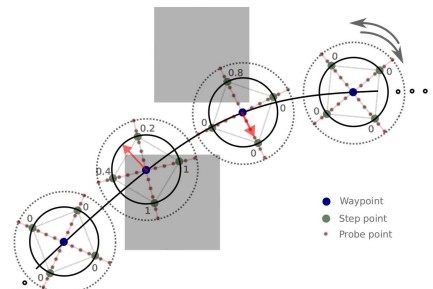

Figure 2: Graphical illustration of Sinkhorn Step with practical considerations. In this point-mass example, we zoom-in one part of the discretized trajectory. The search-direction sets are constructed from randomly rotated 2-cube vertices at each iteration, depicted by the gray arrows and the green points. The gray numbers are the averaged costs over the red probe points in each vertex direction. Note that for clarity, we only visualize an occupancy obstacle cost. The red arrows describe the updates that transport the waypoints gradually out of the obstacles, depending on the (solid inner) polytope circumcircle $\alpha_k$ and (dotted outer) probe circle $\beta_k$.

### 4.3 Batch trajectory optimization

We leverage our Sinkhorn Step to optimize multiple trajectories in parallel, efficiently providing many feasible solutions for multi-modal planning problems. Specifically, we implement MPOT using Py-Torch [37] for vectorization across different motion plans, randomly rotated polytope constructions, and *probe set* cost evaluations. For a problem instance, we consider $N_p$ trajectories of horizon $T$, and thus, the trajectory set $\mathcal{T} = \{\boldsymbol{\tau}_1, \dots, \boldsymbol{\tau}_{N_p}\}$ is the parameter to be optimized. We can flatten the trajectories into the set of $N = N_p \times T$ waypoints. Now, the tensors of search directions and *probe set* $\boldsymbol{D}^P \in \mathbb{R}^{N \times m \times d}$, $\boldsymbol{H}^P \in \mathbb{R}^{N \times m \times h \times d}$ can be efficiently constructed and evaluated by the state cost function $c(\cdot)$, provided that the cost function is implemented with batch-wise processing (e.g., neural network models in PyTorch). Similarly, the model cost term in Eq. (9) can also be evaluated in batch by vectorizing the computation of the second term in Eq. (10).

At each iteration, it is optional to anneal the stepsize $\alpha_k$ and *probe radius* $\beta_k$. Often we do not know the Lipschitz constant $L$ in practice, so the optimal stepsize cannot be computed. Hence, the Sinkhorn Step might oscillate around some local minima. It is an approximation artifact that can be mitigated by reducing the radius of the ball-search over time, gradually changing from an exploratory to an exploitative behavior. Annealing the ball search radius while keeping the number of probe points increases the chance of approximating better ill-conditioned cost structure, e.g., large condition number locally.

To initialize the trajectories, we randomly sample from the discretized GP prior $\mathcal{T}^0 \sim \mathcal{N}(\boldsymbol{\mu}_0, \boldsymbol{K}_0)$, where $\boldsymbol{\mu}_0$ is a constant-velocity, straight-line trajectory from start-to-goal state, and $\boldsymbol{K}_0 \in \mathbb{R}^{(T \times d) \times (T \times d)}$ is a large GP covariance matrix for exploratory initialization [38, 39] (cf. Appendix B). In execution, we select the lowest cost trajectory $\boldsymbol{\tau}^* \in \mathcal{T}^*$. For collecting a trajectory dataset, all collision-free trajectories $\mathcal{T}^*$ are stored along with contextual data, such as occupancy map, goal state,

**Algorithm 1:** Motion Planning via Optimal Transport

---

$\mathcal{T}^0 \sim \mathcal{N}(\boldsymbol{\mu}_0, \boldsymbol{K}_0)$ and $\boldsymbol{n} = \mathbf{1}_N/N$, $\boldsymbol{m} = \mathbf{1}_m/m$

**while** *termination criteria not met* **do**

    (Optional) $\alpha \leftarrow (1-\epsilon)\alpha$, $\beta \leftarrow (1-\epsilon)\beta$          `// Epsilon Annealing for Sinkhorn Step`

    Construct randomly rotated $D^P, H^P$ and compute the cost matrix $\boldsymbol{C}$ as in Eq. (10)

    Perform Sinkhorn Step $\mathcal{T} \leftarrow \mathcal{T} + \mathbf{S}$

**end**

---

etc. See Algorithm 1 for an overview of MPOT. Further discussions on batch trajectory optimization are in Appendix C.

## 5 Experiments

We experimentally evaluate MPOT in PyBullet simulated tasks, which involve high-dimensional state space, multiple objectives, and challenging costs. First, we benchmark our method against strong motion planning baselines in a densely cluttered 2D-point-mass and a 7-DoF robot arm (Panda) environment. Subsequently, we study the convergence of MPOT empirically. Finally, we demonstrate the efficacy of our method on high-dimensional mobile manipulation tasks with TIAGo++. Additional ablation studies on the design choices of MPOT, and gradient approximation capability on a smooth function of Sinkhorn Step w.r.t. different hyperparameter settings are in the Appendix J.

### 5.1 Experimental setup

In all experiments, all planners optimize first-order trajectories with positions and velocities in configuration space. The batch trajectory optimization dimension is $N \times T \times d$, where $d$ is the full-state concatenating position and velocity.

For the *point-mass* environment, we populate 15 square and circle obstacles randomly and uniformly inside x-y limits of $[-10, 10]$, with each obstacle having a radius or width of 2 (cf. Fig. 1). We generate 100 environment-seeds, and for each environment-seed, we randomly sample 10 collision-free pairs of start and goal states, resulting in 1000 planning tasks. We plan each task in parallel 100 trajectories of horizon 64. A trajectory is considered successful if it is collision-free.

For the *Panda* environment, we also generate 100 environment-seeds. Each environment-seed contains randomly sampled 15 obstacle-spheres having a radius of 10cm inside the x-y-z limits of $[[-0.7, 0.7], [-0.7, 0.7], [0.1, 1.]]$, ensuring that the Panda's initial configuration has no collisions (cf. Appendix I). Then, we sample 5 random collision-free (including self-collision-free) target configurations, resulting in 500 planning tasks, and plan in parallel 10 trajectories containing 64 timesteps.

In the last experiment, we design a realistic high-dimensional mobile manipulation task in PyBullet (cf. Fig. 4). The task comprises two parts: the *fetch* part and *place* part; thus, it requires solving two planning problems. Each plan contains 128 timesteps, and we plan a single trajectory for each planner due to the high-computational and memory demands. We generate 20 seeds by randomly spawning the robot in the room, resulting in 20 tasks.

The motion planning costs are the $SE(3)$ goal, obstacle, self-collision, and joint-limit costs. The state dimension (configuration position and velocity) is $d = 4$ for the point-mass experiment, $d = 14$ for the Panda experiment, and $d = 36$ (3 dimensions for the base, 1 for the torso, and 14 for the two arms) for the mobile manipulation experiment. As for polytope settings, we choose a 4-cube for the point-mass case, a 14-othorplex for Panda, and a 36-othorplex for TIAGo++. Further experiment details are in Appendix I.

**Baselines.** We compare MPOT to popular trajectory planners, which are also straightforward to implement and vectorize in PyTorch for a fair comparison (even if the vectorization is not mentioned in their original papers). The chosen baselines are gradient-based planners: CHOMP [7] and GPMP2 (no interpolation) [8]; sampling-based planners: RRT* [6, 10] and its informed version I-RRT* [40], Stochastic Trajectory Optimization for Motion Planning (STOMP) [9], and the recent work Stochastic Gaussian Process Motion Planning (SGPMP) [13]. We implemented all baselines in PyTorch except for RRT* and I-RRT*, which we plan with a loop using CPU.[2] We found that resetting the tree, rather than reusing it, is much faster for generating multiple trajectories; hence, we reset RRT* and I-RRT* when they find their first solution.

---

[2]To the best of our knowledge, vectorization of RRT* is non-trivial and still an open problem.

Table 1: Trajectory generation benchmarks in densely cluttered environments. RRT* and I-RRT* success and collision-free rates depict the maximum achievable values for all planners. S and PL statistics are computed on successful trajectories only.

| | point-mass Experiment | | | | | Panda Experiment | | | | |
|---|---|---|---|---|---|---|---|---|---|---|
| | T[s] | SUC[%] | GOOD[%] | S | PL | T[s] | SUC[%] | GOOD[%] | S | PL |
| RRT* | $43.2 \pm 15.2$ | $100 \pm 0.$ | $100 \pm 0.$ | $0.43 \pm 0.12$ | $23.8 \pm 4.6$ | $186.9 \pm 184.2$ | $100 \pm 0.$ | $73.8 \pm 26.7$ | $0.17 \pm 0.05$ | $7.8 \pm 2.9$ |
| I-RRT* | $43.6 \pm 13.8$ | $100 \pm 0.$ | $100 \pm 0.$ | $0.43 \pm 0.11$ | $23.9 \pm 4.8$ | $184.2 \pm 166.0$ | $100 \pm 0.$ | $74.6 \pm 29.0$ | $0.17 \pm 0.05$ | $7.6 \pm 3.2$ |
| STOMP | $2.2 \pm 0.1$ | $31.4 \pm 13.9$ | $10.5 \pm 25.7$ | $\mathbf{0.01} \pm 0.01$ | $\mathbf{17.0} \pm 1.4$ | $4.3 \pm 0.1$ | $50.8 \pm 28.3$ | $35.3 \pm 42.0$ | $\mathbf{0.01} \pm 0.0$ | $\mathbf{4.5} \pm 0.8$ |
| SGPMP | $6.5 \pm 0.9$ | $98.6 \pm 4.5$ | $\mathbf{74.9} \pm 28.9$ | $0.03 \pm 0.01$ | $18.3 \pm 2.0$ | $5.0 \pm 0.2$ | $67.8 \pm 23.5$ | $58.1 \pm 45.8$ | $\mathbf{0.01} \pm 0.0$ | $4.5 \pm 0.9$ |
| CHOMP | $0.5 \pm 0.1$ | $70.9 \pm 16.7$ | $38.6 \pm 40.7$ | $0.03 \pm 0.0$ | $17.7 \pm 1.7$ | $3.1 \pm 0.3$ | $63.0 \pm 25.5$ | $51.6 \pm 46.2$ | $0.02 \pm 0.0$ | $4.6 \pm 0.8$ |
| GPMP2 | $2.8 \pm 0.1$ | $98.3 \pm 4.9$ | $\mathbf{74.9} \pm 32.1$ | $0.07 \pm 0.05$ | $20.3 \pm 3.1$ | $3.3 \pm 0.2$ | $66.0 \pm 25.2$ | $53.2 \pm 42.3$ | $\mathbf{0.01} \pm 0.0$ | $4.9 \pm 0.8$ |
| **MPOT** | $\mathbf{0.4} \pm 0.0$ | $\mathbf{99.2} \pm 3.1$ | $73.6 \pm 26.7$ | $0.06 \pm 0.03$ | $19.3 \pm 2.3$ | $\mathbf{0.8} \pm 0.1$ | $\mathbf{71.6} \pm 23.2$ | $\mathbf{60.2} \pm 44.4$ | $\mathbf{0.01} \pm 0.01$ | $4.6 \pm 0.9$ |

(a)                                                        (b)

Figure 3: Convergence analysis of MPOT in Panda benchmark. The plots show the cost convergence when applying *step radius* annealing $\epsilon = 0.035$ and without. The plots imply that by following the Sinkhorn Steps, even without annealing, the cost converges exponentially (w.r.t. the update step size shown by the displacement norm). Slower convergence is observed without annealing. The right plots depict the number of iterations for solving the inner OT problem, whose stopping threshold is set at $10^{-5}$. The mean and median of the violin plots are shown in blue and red, respectively. This shows that later iterations require fewer OT iterations, which attributes to the efficiency of MPOT.

**Metrics.** Comparing various aspects among different types of motion planners is challenging. We aim to benchmark the capability of planners to parallelize trajectory optimization under dense environment settings. The metrics are T[s] - *planning time* until convergence; SUC[%] - *success rate* over tasks in an environment-seed, where success means there is at least one successful trajectory found each task; GOOD[%] - success percentage of total parallelized plans in each environment-seed, reflecting the *parallelization quality*; S - *smoothness* measured by the norm of the finite difference of trajectory velocities, averaged over all trajectories and horizons; PL - *path length*.

### 5.2 Benchmarking results

We present the comparative results between MPOT and the baselines in Table 1. While RRT* and I-RRT* achieve perfect results on success criteria, their planning time is dramatically high, which reconfirms the issues of RRT* in narrow passages and high-dimensional settings. Moreover, solutions of RRT* need postprocessing to improve smoothness. For GPMP2, the success rate is comparable but requires computational effort. CHOMP, known for its limitation in narrow passages [41], requiring a small stepsize to work. This parallelization setting requires a larger step size for all trajectories to make meaningful updates, which incurs its inherent instability. With STOMP and SGPMP the comparison is "fairer," as they are both gradient-free methods. However, the sampling variance of STOMP is too restrictive, leading to bad solutions along obstacles near the start and goal configuration. Only SGPMP is comparable in success rate and parallelization quality. Nevertheless, we observe that tuning the proposal distribution variances is difficult in dense environments since they do not consider an obstacle model and cannot sample meaningful "sharp turns", hence requiring small update step size, more samples per evaluation, and longer iterations to optimize.

MPOT achieves better planning time, success rate, and parallelization quality, some with large margins, especially for the Panda experiments, while retaining smoothness due to the GP cost. We observe that MPOT performs particularly well in narrow passages, since waypoints across all trajectories are updated independently, thus avoiding the observed diminishing stepsize issue of the gradient-based planners in parallelization settings. Thanks to the explicit trust region property (cf. Appendix D), it is easier to tune the stepsize since it ensures that the update bound of each waypoint is the polytope convex hull. Notably, the MPOT planning time scales well with dimensionality. As seen in Fig. 3, solving OT is even more rapid at later Sinkhorn Steps; as the waypoints approach local minima, the OT cost matrix becomes more uniform and can be solved with only one or two Sinkhorn iterations.

## 5.3 Mobile manipulation experiment

We design a long-horizon, high-dimensional whole-body mobile manipulation planning task to stress-test our algorithm. This task requires designing many non-convex costs, e.g., signed-distance fields for gradient-based planners. Moreover, the task space is huge while the $SE(3)$ goal is locally small (i.e., the local grasp-pose, while having hyper-redundant mobile robot, meaning the whole-body IK solution may be unreliable); hence, it typically requires long-horizon configuration trajectories and a small update step-size. Notably, the RRTs fail to find a solution, even with a very high time budget of 1000 seconds, signifying the environment's difficulty. These factors also add to the worst performance of GPMP2 in planning time (Table 2). Notably, CHOMP performs worse than GPMP2 and takes more iterations in a cluttered environment in Table 1. However, CHOMP beats GPMP2 in runtime complexity, in this case due to its simpler update rule. STOMP exploration mechanism is restrictive, and we could not tune it to work in this environment. MPOT achieves much better planning times by avoiding the propagation of gradients in a long computational chain while retaining the performance with the efficient Sinkhorn Step, facilitating individual waypoint exploration. However, due to the sparsity of the 36-othorplex ($m = 72$) defining

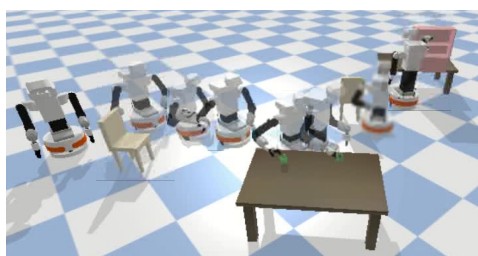

Figure 4: A TIAGo++ robot has to fetch a cup from a table in a room, then put the cup back on the red shelf while avoiding collisions with the chairs.

Table 2: Mobile fetch & place. TF[$s$] depicts the planning time for achieving first successful solution. The average S and PL are evaluated on successful trajectories only. RRT* fails to recover a solution with a very high time budget of 1000 seconds, signifying the environment difficulty.

| | TF[$s$] | SUC[%] | S | PL |
|---|---|---|---|---|
| RRT* | $1000 \pm 0.00$ | 0 | - | - |
| I-RRT* | $1000 \pm 0.00$ | 0 | - | - |
| | | | | |
| STOMP | - | 0 | - | - |
| SGPMP | $27.75 \pm 0.29$ | 25 | $\mathbf{0.010} \pm 0.001$ | $\mathbf{6.69} \pm 0.38$ |
| CHOMP | $16.74 \pm 0.21$ | 40 | $0.015 \pm 0.001$ | $8.60 \pm 0.73$ |
| GPMP2 | $40.11 \pm 0.08$ | 40 | $0.012 \pm 0.015$ | $8.63 \pm 0.53$ |
| **MPOT** | $\mathbf{1.49} \pm 0.02$ | **55** | $0.022 \pm 0.003$ | $10.53 \pm 0.62$ |

the search direction bases in this high-dimensional case, it becomes hard to balance success rate and smoothness when tuning the algorithm, resulting in worse smoothness than the baselines.

**Limitations.** MPOT is backed by experimental evidence that its planning time scales distinctively with high-dimensional tasks in the parallelization setting while optimizing reasonably smooth trajectories. Our experiment does not imply that MPOT should replace prior methods. MPOT has limitations in some aspects. First, the entropic-regularized OT has numerical instabilities when the cost matrix dimension is huge (i.e., huge number of waypoints and vertices). We use log-domain stabilization to mitigate this issue [35, 36]. However, in rare cases, we still observe that the Sinkhorn scaling factors diverge, and MPOT would terminate prematurely. Normalizing the cost matrix, scaling down the cost terms, and slightly increasing the entropy regularization $\lambda$ helps. Second, on the theoretical understanding, we only perform preliminary analysis based on Assumption 2 to connect directional-direct search literature. Analyzing Sinkhorn Steps in other conditions for better understanding, e.g., Sinkhorn Step gradient approximation analysis with arbitrary $\lambda > 0$, Sinkhorn Step on convex functions for sharper complexity bounds, etc., is desirable. Third, learning motion priors [13, 24] can naturally complement MPOT to provide even better initializations, as currently, we only use GP priors to provide random initial smooth trajectories.

## 6 Conclusions and Broader Impacts

We presented MPOT—a gradient-free and efficient batch motion planner that optimizes multiple high-dimensional trajectories over non-convex objectives. In particular, we proposed the Sinkhorn Step—a zero-order batch update rule parameterized by a local optimal transport plan with a nice property of cost-agnostic step bound, effectively updating waypoints across trajectories independently. We demonstrated that in practice, our method converges, scales very well to high-dimensional tasks, and provides practically smooth plans. This work opens multiple exciting research questions, such as investigating further polytope families that can be applied for scaling up to even more high-dimensional settings, conditional batch updates, or different strategies for adapting the step-size. Furthermore, while classical motion planning considers single planning instance for each task, which under-utilizes the modern GPU capability, this work encourages future work that benefits from vectorization in the algorithmic choices, providing multiple plans and covering several modes, leading to execution robustness or even for dataset collection for downstream learning tasks. At last, we foresee potential applications of Sinkhorn Step to sampling methods or variational inference.

## Acknowledgments and Disclosure of Funding

An T. Le was supported by the German Research Foundation project METRIC4IMITATION (PE 2315/11-1). Georgia Chalvatzaki was supported by the German Research Foundation (DFG) Emmy Noether Programme (CH 2676/1-1). We also gratefully acknowledge Alexander Lambert for his implementation of the Gaussian Process prior; Pascal Klink, Joe Watson, João Carvalho, and Julen Urain for the insightful and fruitful discussions; Snehal Jauhri, Puze Liu, and João Carvalho for their help in setting up the TIAGo++ environments.

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

# A Theoretical Analysis & Proofs

We investigate the proposed Sinkhorn Step (Definition 2 in Section 3.2) properties for a non-convex and smooth objective function (Assumption 1). Our preliminary analysis performs at arbitrary iteration $k > 0$ and depends on Assumption 1 and Assumption 2 stated in Section 3.

First, we state a proof sketch of Proposition 1.

**Proposition 1.** $\forall P \in \mathcal{P}$, $D^P$ forms a positive spanning set.

*Proof.* Observe that by construction of $d$-dimensional regular polytope $P \in \mathcal{P}$, the convex hull of its vertex set $\mathcal{V}^P$

$$\mathrm{conv}(\mathcal{V}^P) = \left\{ \sum_i w_i \boldsymbol{v}_i \mid \boldsymbol{v}_i \in \mathcal{V}^P, \sum_i w_i = 1, \, w_i > 0, \, \forall i \right\}$$

has $\dim(\mathrm{conv}(\mathcal{V}^P)) = d$ dimensions. Hence, trivially, the conic hull of $D^P$ positively spans $\mathbb{R}^d$. □

Now, we can investigate the quality of $D^P$ in the following lemma.

**Lemma 1.** *For any* $\boldsymbol{a} \in \mathbb{R}^d, \boldsymbol{a} \neq \boldsymbol{0}$, $\exists \boldsymbol{d} \in D^P$ *such that*

$$\langle \boldsymbol{a}, \boldsymbol{d} \rangle \geq \mu_P \|\boldsymbol{a}\|, \, 0 \leq \mu_p \leq 1$$

*where* $\mu_P = 1/\sqrt{d(d+1)}$ *for* $P = d$-*simplex,* $\mu_P = 1/\sqrt{d}$ *for* $P = d$-*orthoplex, and* $\mu_P = 1/\sqrt{2}$ *for* $P = d$-*cube.*

*Proof.* From Proposition 1, $D^P$ is a positive spanning set, then for any $\boldsymbol{a} \in \mathbb{R}^d$, $\exists \boldsymbol{d} \in D^P$ such that $\langle \boldsymbol{a}, \boldsymbol{d} \rangle > 0$ (Theorem 2.6, [28]). This property results in the positive cosine measure of $D^P$(Proposition 7, [42])

$$1 \geq \mu_P := \min_{\boldsymbol{0} \neq \boldsymbol{a} \in \mathbb{R}^d} \max_{\boldsymbol{d} \in D^P} \frac{\langle \boldsymbol{a}, \boldsymbol{d} \rangle}{\|\boldsymbol{a}\| \|\boldsymbol{d}\|} > 0 \tag{11}$$

Equivalently, $\mu_P$ is the largest scalar such that $\langle \boldsymbol{a}, \boldsymbol{d} \rangle \geq \mu_P \|\boldsymbol{a}\| \|\boldsymbol{d}\| = \mu_P \|\boldsymbol{a}\|, \, \boldsymbol{d} \in D^P$.

Next, due to the symmetry of the regular polytope family $\mathcal{P}$, there exists an inscribing hypersphere $S_r^{d-1}$ with radius $r$ for any $P \in \mathcal{P}$ [29]. For $\mathcal{P}$, the tangent points of the inscribing hypersphere to the facets are also the centroid of the facets. Then, the centroid vectors pointing from the origin towards these tangent points form equal angles to all nearby vertex vectors. Thus, the cosine measure attains its saddle points Eq. (11) at these centroid vectors having the value

$$\mu_P = \frac{r}{R}$$

with $R = 1$ is the radius of the circumscribed unit hypersphere. The inradius $r$ for $d$-simplex, $d$-orthoplex, and $d$-cube are $1/\sqrt{d(d+1)}, 1/\sqrt{d}, 1/\sqrt{2}$, respectively [29]. □

This lemma has a straightforward geometric implication - for every $\boldsymbol{v} \neq \boldsymbol{0}, \boldsymbol{v} \in \mathbb{R}^d$, there exists a search direction $\boldsymbol{d} \in D^P$ such that the cosine angle between these vectors is acute (i.e., $\mu_P > 0$). Then, if we consider the negative gradient vector, which is unknown, there exists a direction in $D^P$ that approximates it well with $\mu_P$ being the quality metric (i.e., larger $\mu_P$ is better). The values of $\mu_P$ for each polytope type also confirm the intuition that, for $d$-cube with an exponential number of vertices $m = 2^d$ has a constant cosine measure, while the cosine measure of $d$-simplex having $m = d + 1$ vertices scales $O(1/d)$ with dimension. Now, we state the key lemma used to prove the main property of Sinkhorn Step.

**Lemma 2** (Key lemma). *If Assumption 1 and Assumption 2 holds, then* $\forall \boldsymbol{x}_k \in X_k, \forall k > 0$

$$f(\boldsymbol{x}_{k+1}) \leq f(\boldsymbol{x}_k) - \mu_P \alpha_k \|\nabla f(\boldsymbol{x}_k)\| + \frac{L}{2} \alpha_k^2 \tag{12}$$

*Proof.* If the Assumption 2 holds, by Proposition 4.1 in [17], the OT solution $\boldsymbol{W}_\lambda \to \boldsymbol{W}_0$ converges to the optimal solution with maximum entropy in the set of solutions of the original problem

$$\min_{\boldsymbol{W} \in U(\mathbf{1}_n/n, \mathbf{1}_n/n)} \langle \boldsymbol{W}, \boldsymbol{C} \rangle.$$

Moreover, Birkhoff doubly stochastic matrix theorem [43] states that the set of extremal points of $U(\mathbf{1}_n/n, \mathbf{1}_n/n)$ is equal to the set of permutation matrices, and the fundamental theorem of linear programming (Theorem 2.7 in [44]) states that the minimum of a linear objective in a finite non-empty polytope is reached at a vertex or a face of the polytope (i.e., the feasible space of the linear program), leading to the following two cases.

- **Case 1**: $\boldsymbol{W}_\lambda/n \to \boldsymbol{W}_0/n$ converges to a permutation matrix representing the bijective mapping between the optimizing points and the polytope vertices. There exists a vertex evaluation permutation forming the cost matrix such that, the update step $\boldsymbol{s}_k$ is a descending step for each optimizing point

$$\forall \boldsymbol{x}_k \in X_k, \ \boldsymbol{s}_k = \alpha_k \frac{1}{n} \boldsymbol{w}_0^* \boldsymbol{D}^P = \operatorname*{argmin}_{\boldsymbol{d} \in D^P} \{ f(\boldsymbol{x}_k + \alpha_k \boldsymbol{d}) \} \tag{13}$$

  with $\boldsymbol{w}_0^*$ as a row in $\boldsymbol{W}_0^*$, then $\boldsymbol{w}_0^*/n$ is a one-hot vector. Let $\boldsymbol{a} = -\nabla f(\boldsymbol{x}_k)$, $\boldsymbol{s}_k$ is a descending step $f(\boldsymbol{x}_k + \boldsymbol{s}_k) \leq f(\boldsymbol{x}_k)$, then, by Lemma 1, $\langle \nabla f(\boldsymbol{x}_k), \boldsymbol{s}_k \rangle \leq -\alpha_k \mu_P \|\nabla f(\boldsymbol{x}_k)\|$.

- **Case 2**: $\boldsymbol{W}_\lambda/n \to \boldsymbol{W}_0/n$ converges to a linear interpolation between the permutation matrices defining the neighboring vertices of the polytope. In this case, there are infinite solutions as the linear interpolation between the two bijective maps. There still exists a vertex evaluation permutation forming the cost matrix such that, the update step $\boldsymbol{s}_k$ is the linear interpolation of multiple tied descending steps for each optimizing point, with $\boldsymbol{s}_k = \sum_i b_i \boldsymbol{d}_i$, $\sum_i b_i = 1$, $b_i \geq 0$, $\boldsymbol{d}_i = \operatorname{argmin}_{\boldsymbol{d} \in D^P} \{ f(\boldsymbol{x}_k + \alpha_k \boldsymbol{d}) \}$. Following the argument of Case 1, since $\boldsymbol{s}_k$ is the linear interpolation of descending steps, we also conclude that $\langle \nabla f(\boldsymbol{x}_k), \boldsymbol{s}_k \rangle = \sum_i b_i \langle \nabla f(\boldsymbol{x}_k), \boldsymbol{d}_i \rangle \leq -\sum_i b_i \alpha_k \mu_P \|\nabla f(\boldsymbol{x}_k)\| = -\alpha_k \mu_P \|\nabla f(\boldsymbol{x}_k)\|$.

Finally, starting the L-smooth property of $f$, we can write

$$\forall \boldsymbol{x} \in X, \ \forall k > 0, \ f(\boldsymbol{x}_{k+1}) = f(\boldsymbol{x}_k + \boldsymbol{s}_k) \leq f(\boldsymbol{x}_k) + \langle \nabla f(\boldsymbol{x}_k), \boldsymbol{s}_k \rangle + \frac{L}{2} \|\boldsymbol{s}_k\|^2$$
$$\leq f(\boldsymbol{x}_k) - \mu_P \alpha_k \|\nabla f(\boldsymbol{x}_k)\| + \frac{L}{2} \alpha_k^2 \tag{14}$$

recalling that $\|\boldsymbol{d}\| = 1, \forall \boldsymbol{d} \in D^P$. $\qquad \square$

If the sufficient decrease condition does not hold $f(\boldsymbol{x}_k) - f(\boldsymbol{x}_{k+1}) < c\alpha_k^2$ with some $c > 0$, then the iteration is deemed unsuccessful. In fact, Lemma 2 is similar to (Lemma 10, [42]), which states that the gradients for these unsuccessful iterations are bounded above by a scalar multiplied with the stepsize. We can see this by rewriting Eq. (12) as

$$\|\nabla f(\boldsymbol{x}_k)\| \leq \frac{1}{\mu_P} \left( \frac{f(\boldsymbol{x}_k) - f(\boldsymbol{x}_{k+1})}{\alpha_k} + \frac{L}{2} \alpha_k \right)$$
$$< \frac{1}{\mu_P} \left( c + \frac{L}{2} \right) \alpha_k.$$

We can implement a check if the sufficient decrease condition holds for ensuring monotonicity in each iteration, as a variant of the Sinkhorn Step.

Lemma 2 also enables analyzing each optimizing point separately, and hence we can state the following main theorem separately for each $\boldsymbol{x}_k \in X_k$.

**Theorem 1** (Main result). *If Assumption 1 and Assumption 2 holds at each iteration and the stepsize is sufficiently small $\alpha_k = \alpha$ with $0 < \alpha < 2\mu_P \epsilon / L$, then with a sufficient number of iteration*

$$K \geq k(\epsilon) := \frac{f(\boldsymbol{x}_0) - f_*}{(\mu_P \epsilon - \frac{L\alpha}{2})\alpha} - 1,$$

*we have* $\min_{0 \leq k \leq K} \|\nabla f(\boldsymbol{x}_k)\| \leq \epsilon, \forall \boldsymbol{x}_k \in X_k$.

*Proof.* We attempt the proof by contradiction, thus we assume $\|\nabla f(\boldsymbol{x}_k)\| > \epsilon$ for all $k \leq k(\epsilon)$. From Lemma 2, we have $\forall \boldsymbol{x}_k \in X_k$, $\forall k > 0$

$$f(\boldsymbol{x}_{k+1}) \leq f(\boldsymbol{x}_k) - \mu_P \alpha \|\nabla f(\boldsymbol{x}_k)\| + \frac{L}{2}\alpha^2.$$

From Assumption 1, the objective is bounded below $f_* \leq f(\boldsymbol{x})$. Hence, we can write

$$
\begin{aligned}
f_* \leq f(\boldsymbol{x}_{K+1}) &< f(\boldsymbol{x}_K) - \mu_P \alpha \|\nabla f(\boldsymbol{x}_K)\| + \frac{L}{2}\alpha^2 \\
&\leq f(\boldsymbol{x}_{K-1}) - \mu_P \alpha(\|\nabla f(\boldsymbol{x}_K)\| + \|\nabla f(\boldsymbol{x}_{K-1})\|) + 2\frac{L}{2}\alpha^2 \\
&\leq f(\boldsymbol{x}_0) - \mu_P \alpha \sum_{k=0}^{K} \|\nabla f(\boldsymbol{x}_k)\| + (K+1)\frac{L}{2}\alpha^2 \\
&\leq f(\boldsymbol{x}_0) - (K+1)\mu_P \alpha \epsilon + (K+1)\frac{L}{2}\alpha^2 \\
&\leq f(\boldsymbol{x}_0) - (K+1)(\mu_P \alpha \epsilon - \frac{L}{2}\alpha^2) \\
&\leq f(\boldsymbol{x}_0) - (f(\boldsymbol{x}_0) - f_*) \\
&= f_*
\end{aligned}
\tag{15}
$$

by applying recursively Lemma 2 and the iteration lower bound at the second last line, which is a contradiction $f_* \leq f_*$. Hence, $\|\nabla f(\boldsymbol{x}_k)\| \leq \epsilon$ for some $k \leq k(\epsilon)$. $\qquad\square$

If $L$ is known, we can compute the optimal stepsize $\alpha = \mu_P \epsilon / L$. Then, the complexity bound is $k(\epsilon) = \frac{2L(f(\boldsymbol{x}_0) - f_*)}{\mu_P^2 \epsilon^2} - 1$. Note that Theorem 1 only guarantees the gradient of some points in the sequence of Sinkhorn Steps will be arbitrarily small. If in practice, we implement the sufficient decreasing condition $f(\boldsymbol{x}_k) - f(\boldsymbol{x}_{k+1}) \geq c\alpha_k^2$, then $f(\boldsymbol{x}_K) \leq f(\boldsymbol{x}_i)$, $\|\nabla f(\boldsymbol{x}_i)\| \leq \epsilon$ holds. However, this sufficient decrease check may waste some iterations and worsen the performance. We show in the experiments that the algorithm exhibits convergence behavior without this condition checking. Finally, we remark on the complexity bounds when using different polytope types for Sinkhorn Step under Assumption 1 and Assumption 2, by substituting $\mu_P$ according to Lemma 1.

**Remark 1.** *By Theorem 1, with the optimal stepsize $\alpha = \mu_P \epsilon / L$, the complexity bounds for $d$-simplex, $d$-orthoplex and $d$-cube are $O(d^2/\epsilon^2)$, $O(d/\epsilon^2)$, and $O(1/\epsilon^2)$, respectively.*

The optimal stepsize with $d$-simplex reports the same complexity $O(d^2/\epsilon^2)$ as the best-known bound for directional-direct search [45]. Within the directional-direct search scope, $d$-cube reports the new best-known complexity bound $O(1/\epsilon^2)$, which is independent of dimension $d$ since the number of search directions is also increased exponentially with dimension. However, in practice, solving a batch update with $d$-cube for each iteration is expensive since now the column-size of the cost matrix is $2^d$.

## B   Gaussian Process Trajectory Prior

To provide a trajectory prior with tunable time-correlated covariance for trajectory optimization, either as initialization prior or as cost, we introduce a prior for continuous-time trajectories using a GP [46, 47, 8]: $\boldsymbol{\tau} \sim \mathcal{GP}(\boldsymbol{\mu}(t), \boldsymbol{K}(t, t'))$, with mean function $\boldsymbol{\mu}$ and covariance function $\boldsymbol{K}$. As described in [47, 48, 13], a GP prior can be constructed from a linear time-varying stochastic differential equation

$$\dot{\boldsymbol{x}} = \boldsymbol{A}(t)\boldsymbol{x}(t) + \boldsymbol{u}(t) + \boldsymbol{F}(t)\mathbf{w}(t) \tag{16}$$

with $\boldsymbol{u}(t)$ the control input, $\boldsymbol{A}(t)$ and $\boldsymbol{F}(t)$ the time-varying system matrices, and $\mathbf{w}(t)$ a disturbance following the white-noise process $\mathbf{w}(t) \sim \mathcal{GP}(\mathbf{0}, \boldsymbol{Q}_c \delta(t - t'))$, where $\boldsymbol{Q}_c \succ 0$ is the power-spectral density matrix. With a chosen discretization time $\Delta t$, the continuous-time GP can be parameterized by a mean vector of Markovian support states $\boldsymbol{\mu} = [\boldsymbol{\mu}(0), ..., \boldsymbol{\mu}(T)]^\intercal$ and covariance matrix $\boldsymbol{K} = [\boldsymbol{K}(i,j)]_{ij, 0 \leq i,j \leq T}$, $\boldsymbol{K}(i,j) \in \mathbb{R}^{d \times d}$, resulting in a multivariate Gaussian $q(\boldsymbol{\tau}) = \mathcal{N}(\boldsymbol{\mu}, \boldsymbol{K})$.

The inverse of the covariance matrix has a sparse structure $\boldsymbol{K}^{-1} = \boldsymbol{D}^{\intercal}\boldsymbol{Q}^{-1}\boldsymbol{D}$ (Theorem 1 in [47]) with

$$
\boldsymbol{D} = \begin{bmatrix} \boldsymbol{I} & & & & \\ -\boldsymbol{\Phi}_{1,0} & \boldsymbol{I} & & & \\ & & \cdots & & \\ & & & \boldsymbol{I} & \boldsymbol{0} \\ & & & -\boldsymbol{\Phi}_{T,T-1} & \boldsymbol{I} \\ & & & \boldsymbol{0} & \boldsymbol{I} \end{bmatrix}, \tag{17}
$$

and the block diagonal time-correlated noise matrix $\boldsymbol{Q}^{-1} = \mathrm{diag}(\boldsymbol{\Sigma}_s^{-1}, \boldsymbol{Q}_{0,1}^{-1}, \ldots, \boldsymbol{Q}_{T-1,T}^{-1}, \boldsymbol{\Sigma}_g^{-1})$. Here, $\boldsymbol{\Phi}_{t,t+1}$ is the state transition matrix, $\boldsymbol{Q}_{t,t+1}$ the covariance between time step $t$ and $t+1$, and $\boldsymbol{\Sigma}_s, \boldsymbol{\Sigma}_g$ are the chosen covariance of the start and goal states. In this work, we mainly consider the constant-velocity prior (i.e., white-noise-on-acceleration model $\ddot{\mathbf{x}}(t) = \mathbf{w}(t)$), which can approximately represent a wide range of systems such as point-mass dynamics, gravity-compensated robotics arms [8], differential drive [47], etc., while enjoying its sparse structure for computation efficiency. As used in our paper, this constant-velocity prior can be constructed from Eq. (16) with the Markovian state representation $\boldsymbol{x} = [\mathbf{x}, \dot{\mathbf{x}}] \in \mathbb{R}^d$ and

$$
\boldsymbol{A}(t) = \begin{bmatrix} \boldsymbol{0} & \boldsymbol{I}_{d/2} \\ \boldsymbol{0} & \boldsymbol{0} \end{bmatrix}, \quad \boldsymbol{u} = \boldsymbol{0}, \quad \boldsymbol{F}(t) = \begin{bmatrix} \boldsymbol{0} \\ \boldsymbol{I}_{d/2} \end{bmatrix} \tag{18}
$$

Then, following [47, 8], the state transition and covariance matrix are

$$
\boldsymbol{\Phi}_{t,t+1} = \begin{bmatrix} \boldsymbol{I}_{d/2} & \Delta t \boldsymbol{I}_{d/2} \\ \boldsymbol{0} & \boldsymbol{I}_{d/2} \end{bmatrix}, \quad \boldsymbol{Q}_{t,t+1} = \begin{bmatrix} \frac{1}{3}\Delta t^3 \boldsymbol{Q}_c & \frac{1}{2}\Delta t^2 \boldsymbol{Q}_c \\ \frac{1}{2}\Delta t^2 \boldsymbol{Q}_c & \Delta t \boldsymbol{Q}_c \end{bmatrix} \tag{19}
$$

with the inverse

$$
\boldsymbol{Q}_{t,t+1}^{-1} = \begin{bmatrix} 12\Delta t^{-3}\Delta t^3 \boldsymbol{Q}_c^{-1} & -6\Delta t^{-2}\Delta t^3 \boldsymbol{Q}_c^{-1} \\ -6\Delta t^{-2}\Delta t^3 \boldsymbol{Q}_c^{-1} & 4\Delta t^{-1} \boldsymbol{Q}_c^{-1} \end{bmatrix}, \tag{20}
$$

which are used to compute $\boldsymbol{K}$.

Consider priors on start state $q_s(\boldsymbol{x}) = \mathcal{N}(\boldsymbol{\mu}_s, \boldsymbol{\Sigma}_s)$ and goal state $q_g(\boldsymbol{x}) = \mathcal{N}(\boldsymbol{\mu}_g, \boldsymbol{\Sigma}_g)$, the GP prior for discretized trajectory can be factored as follows [47, 8]

$$
\begin{aligned}
q_F(\boldsymbol{\tau}) &\propto \exp\left(-\frac{1}{2}\|\boldsymbol{\tau} - \boldsymbol{\mu}\|_{\boldsymbol{K}^{-1}}^2\right) \\
&\propto q_s(\boldsymbol{x}_0)\, q_g(\boldsymbol{x}_T) \prod_{t=0}^{T-1} q_t(\boldsymbol{x}_t, \boldsymbol{x}_{t+1}),
\end{aligned} \tag{21}
$$

where each binary GP-factor is defined

$$
q_t(\boldsymbol{x}_t, \boldsymbol{x}_{t+1}) = \exp\left\{-\frac{1}{2}\|\boldsymbol{\Phi}_{t,t+1}(\boldsymbol{x}_t - \boldsymbol{\mu}_t) - (\boldsymbol{x}_{t+1} - \boldsymbol{\mu}_{t+1})\|_{\boldsymbol{Q}_{t,t+1}^{-1}}^2\right\}. \tag{22}
$$

In the main paper, we use this constant-velocity GP formulation to sample initial trajectories. The initialization GP is parameterized by the constant-velocity straight line $\boldsymbol{\mu}_0$ connecting a start configuration $\boldsymbol{\mu}_s$ to a goal configuration $\boldsymbol{\mu}_g$, having moderately high covariance $\boldsymbol{K}_0$. For using this GP as the cost, we set the zero-mean $\boldsymbol{\mu} = \boldsymbol{0}$ to describe the uncontrolled trajectory distribution. The conditioning $q_g(\boldsymbol{x}_T)$ of the final waypoint to the goal configuration $\boldsymbol{\mu}_g$ is optional (e.g., when the goal configuration solution from inverse kinematics is sub-optimal), and we typically use the $SE(3)$ goal cost.

## C   Additional Discussions Of Batch Trajectory Optimization

**Direct implications of batch trajectory optimization**. MPOT can be used as a strong oracle for collecting datasets due to the solution diversity covering various modes, capturing homotopy classes of the tasks and their associated contexts. For direct execution, with high variance initialization, an abundance of solutions vastly increases the probability of discovering good local minima, which we can select the best solution according to some criteria, e.g., collision avoidance, smoothness, model consistency, etc.

**Solution diversity of MPOT**. Batch trajectory optimization can serve as a strong oracle for collecting datasets or striving to discover a global optimal trajectory for execution. Three main interplaying factors contribute to the solution diversity, hence discovering better solution modes. They are

- the step radius $\alpha_k$ annealing scheme,
- the variances of GP prior initialization,
- the number of plans in a batch.

Additional sampling mechanism that promotes diversity, such as Stein Variational Gradient Descent (SVGD) [49] can be straightforwardly integrated into the trajectory optimization problem [23]. This is considered in the future version of this paper to integrate the SVGD update rule with the Sinkhorn Step (i.e., using the Sinkhorn Step to approximate the score function) for even more diverse trajectory planning.

**Extension to optimizing batch of different trajectory horizons**. Currently, for vectorizing the update of all waypoints across the batch of trajectories, we flatten the batch and horizon dimensions and apply the Sinkhorn Step. After optimization, we reshape the tensor to the original shape. Notice that what glues the waypoints in the same trajectory together after optimization is the log of the Gaussian Process as the model cost, which promotes smoothness and model consistency. Given this pretext, in case of a batch of different horizon trajectories, we address this case by setting maximum horizon $T_{\max}$ and padding with zeros for those trajectories having $T < T_{\max}$. Then, we also set zeros for all rows corresponding to these padded points in the cost matrix $\mathbf{C}^{T_{\max} \times m}$. The padded points are ignored after the barycentric projection. Another way is to maintain an index list of start and end indices of trajectories after flattening, then the cost computation also depends on this index list. Finally, the trajectories with different horizons can be extracted based on the index list. Intuitively, we just need to manipulate cost entries to dictate the behavior of waypoints.

# D   Explicit Trust Region Of The Sinkhorn Step

In trajectory optimization, it is crucial to bound the trajectory update at every iteration to be close to the previous one for stability, and so that the updated parameter remains within the region where the linear approximations are valid. Given $\mathcal{F}(\cdot) : \mathbb{R}^{T \times d} \to \mathbb{R}$ to be the planning cost functional, prior works [7, 50, 8] apply a first-order Taylor expansion at the current parameter $\boldsymbol{\tau}_k$, while adding a regularization norm

$$\Delta \boldsymbol{\tau}^* = \operatorname*{argmin} \left\{ \mathcal{F}(\boldsymbol{\tau}_k) + \nabla \mathcal{F}(\boldsymbol{\tau}_k) \Delta \boldsymbol{\tau} + \frac{\beta}{2} \left\| \Delta \boldsymbol{\tau}_k \right\|_{\boldsymbol{M}} \right\}, \tag{23}$$

resulting in the following update rule by differentiating the right-hand side w.r.t. $\Delta \boldsymbol{\tau}$ and setting it to zero

$$\boldsymbol{\tau}_{k+1} = \boldsymbol{\tau}_k + \Delta \boldsymbol{\tau}^* = \boldsymbol{\tau}_k - \frac{1}{\beta} \boldsymbol{M}^{-1} \nabla \mathcal{F}(\boldsymbol{\tau}_k). \tag{24}$$

The metric $\boldsymbol{M}$ depends on the conditioning prior. Ratliff et al. [7] propose $\boldsymbol{M}$ to be the finite difference matrix, constraining the update to stay in the region of smooth trajectories (i.e., low-magnitude trajectory derivatives). Mukadam et al. [8] use the metric $\boldsymbol{M} = \boldsymbol{K}$ derived from a GP prior, also enforcing the dynamics constraint. It is well-known that solving for $\Delta \boldsymbol{\tau}$ in Eq. (23) is equivalent to minimizing the linear approximation within the ball of radius defined by the third term (i.e., the regularization norm) [51]. Hence, these mechanisms can be interpreted as implicitly shaping the *trust region* - biasing perturbation region by the prior, connecting the prior to the weighting matrix $\boldsymbol{M}$ in the update rule.

In contrast, the Sinkhorn Step approaches the *trust region* problem with a gradient-free perspective and provides a novel way to explicitly constrain the parameter updates inside a trust region defined by the regular polytope, without relying on Taylor expansions, where cost functional derivatives are not always available in practice (e.g., planning with only occupancy maps, planning through contacts). In this work, the bound on the trajectory update by the Sinkhorn Step is straightforward

$$
\begin{aligned}
\left\| \boldsymbol{\tau}_{k+1} - \boldsymbol{\tau}_k \right\| &= \left\| \alpha_k \operatorname{diag}(\boldsymbol{n})^{-1} \boldsymbol{W}_\lambda^* \boldsymbol{D}^P \right\| \\
&\leq \sum_{t=1}^{T} \left\| \alpha_k \frac{1}{n} \boldsymbol{w}_\lambda^* \boldsymbol{D}^P \right\| \\
&\leq \sum_{t=1}^{T} \left\| \alpha_k \boldsymbol{d}^* \right\| \leq T \alpha_k
\end{aligned}
\tag{25}
$$

**Algorithm 2:** Stabilized Sinkhorn Algorithm

---

$(\boldsymbol{a}^0, \boldsymbol{b}^0) \leftarrow (\mathbf{0}_T, \mathbf{0}_m)$, $(\tilde{\boldsymbol{u}}^0, \tilde{\boldsymbol{v}}^0) \leftarrow (\mathbf{1}_T, \mathbf{1}_m)$, $M = 10^3$.
Compute stabilized kernel $\boldsymbol{P}^0$ using Eq. (29).
**while** *termination criteria not met* **do**
    // Sinkhorn iteration
    $\tilde{\boldsymbol{u}}^{i+1} = \boldsymbol{n}/(\boldsymbol{P}^i \tilde{\boldsymbol{v}}^i), \quad \tilde{\boldsymbol{v}}^{i+1} = \boldsymbol{m}/(\boldsymbol{P}^{i\mathsf{T}} \tilde{\boldsymbol{u}}^{i+1}).$
    // Check for numerical instabilities
    **if** $\left\| \tilde{\boldsymbol{u}}^i \right\|_\infty < M \vee \left\| \tilde{\boldsymbol{v}}^i \right\|_\infty < M$ **then**
        // Absorption.
        $(\boldsymbol{a}^i, \boldsymbol{b}^i) \leftarrow (\boldsymbol{a}^i + \lambda \log(\tilde{\boldsymbol{u}}^i), \; \boldsymbol{b}^i + \lambda \log(\tilde{\boldsymbol{v}}^i)).$
        Compute stabilized kernel $\boldsymbol{P}^i$ using Eq. (29).
        $(\tilde{\boldsymbol{u}}^i, \tilde{\boldsymbol{v}}^i) \leftarrow (\mathbf{1}_T, \mathbf{1}_m).$
    **end**
**end**
Return $\boldsymbol{W}^*_\lambda = \mathrm{diag}(\tilde{\boldsymbol{u}}^*)\boldsymbol{P}^*\mathrm{diag}(\tilde{\boldsymbol{v}}^*).$

---

resulting from $\boldsymbol{D}^P$ being a regular polytope inscribing the $(d-1)$-unit hypersphere. In practice, one could scale the polytope in different directions by multiplying with $\boldsymbol{M}$ induced by priors, and, hence, shape the trust region in a similar fashion. Note that the bound in Eq. (25) does not depend on the local cost information.

For completeness of discussion, in sampling-based trajectory optimization, the regularization norm is related to the variance of the proposal distribution. The trajectory candidates are sampled from the proposal distribution and evaluated using the Model-Predictive Path Integral (MPPI) update rule [52]. For example, Kalakrishnan et al. [9] construct the variance matrix similarly to the finite difference matrix, resulting in a sampling distribution with low variance at the tails and high variance at the center. Recently, Urain et al. [13] propose using the same GP prior variance as in [8] to sample trajectory candidates for updates, leveraging them for tuning variance across timesteps.

## E  The Log-Domain Stabilization Sinkhorn Algorithm

Following Proposition 4.1 in [17], for sufficiently small regularization $\lambda$, the approximate solution from the entropic-regularized OT problem

$$\boldsymbol{W}^*_\lambda = \operatorname{argmin} \mathrm{OT}_\lambda(\boldsymbol{n}, \boldsymbol{m})$$

approaches the true optimal plan

$$\boldsymbol{W}^* = \operatorname*{argmin}_{\boldsymbol{W} \in U(\boldsymbol{n}, \boldsymbol{m})} \langle \boldsymbol{C}, \boldsymbol{W} \rangle.$$

However, small $\lambda$ incurs numerical instability for a high-dimensional cost matrix, which is usually the case for our case of batch trajectory optimization. Too high $\lambda$, which leads to "blurry" plans, also harms the MPOT performance. Hence, we utilize the log-domain stabilization for the Sinkhorn algorithm.

We provide a brief discussion of this log-domain stabilization. For a full treatment of the theoretical derivations, we refer to [35, 36]. First, with the marginals $\boldsymbol{n} \in \Sigma_T$, $\boldsymbol{m} \in \Sigma_m$ and the exponentiated kernel matrix $\boldsymbol{P} = \exp(-\boldsymbol{C}/\lambda)$, the Sinkhorn algorithm aims to find a pair of scaling factors $\boldsymbol{u} \in \mathbb{R}^T_+$, $\boldsymbol{v} \in \mathbb{R}^m_+$ such that

$$\boldsymbol{u} \odot \boldsymbol{P}\boldsymbol{v} = \boldsymbol{n}, \quad \boldsymbol{v} \odot \boldsymbol{P}^\mathsf{T}\boldsymbol{u} = \boldsymbol{m}, \tag{26}$$

where $\odot$ is the element-wise multiplication (i.e., the Hadamard product). From a typical one vector initialization $\boldsymbol{v}^0 = \mathbf{1}_m$, the Sinkhorn algorithm performs a sequence of (primal) update rules

$$\boldsymbol{u}^{i+1} = \frac{\boldsymbol{n}}{\boldsymbol{P}\boldsymbol{v}^i}, \quad \boldsymbol{v}^{i+1} = \frac{\boldsymbol{m}}{\boldsymbol{P}^\mathsf{T}\boldsymbol{u}^{i+1}}, \tag{27}$$

leading to convergence of the scaling factors $\boldsymbol{u}^*, \boldsymbol{v}^*$ [53]. Then, the optimal transport plan can be computed by $\boldsymbol{W}^*_\lambda = \mathrm{diag}(\boldsymbol{u}^*)\boldsymbol{P}\mathrm{diag}(\boldsymbol{v}^*)$.

For small values of $\lambda$, the entries of $\boldsymbol{P}, \boldsymbol{u}, \boldsymbol{v}$ become either very small or very large, thus being susceptible to numerical problems (e.g., floating point underflow and overflow). To mitigate this

issue, at an iteration $i$, Chizat et al. [35] suggests a redundant parameterization of the scaling factors as

$$\boldsymbol{u}^i = \tilde{\boldsymbol{u}}^i \odot \exp(\boldsymbol{a}^i/\lambda), \quad \boldsymbol{v}^i = \tilde{\boldsymbol{v}}^i \odot \exp(\boldsymbol{b}^i/\lambda), \tag{28}$$

with the purpose of keeping $\tilde{\boldsymbol{u}}, \tilde{\boldsymbol{v}}$ bounded, while absorbing extreme values of $\boldsymbol{u}, \boldsymbol{v}$ into the log-domain via redundant vectors $\boldsymbol{a}, \boldsymbol{b}$. The kernel matrix $\boldsymbol{P}^i$ is also stabilized, having elements being modified as

$$\boldsymbol{P}_{tj}^i = \exp\left((\boldsymbol{a}_t^i + \boldsymbol{b}_j^i - \boldsymbol{C}_{tj})/\lambda\right), \tag{29}$$

such that large values in $\boldsymbol{a}, \boldsymbol{b}$ and $\boldsymbol{C}$ cancel out before the exponentiation, which is crucial for small $\lambda$. With these ingredients, we state the log-domain stabilization Sinkhorn algorithm in Algorithm 2. Note that Algorithm 2 is mathematically equivalent to the original Sinkhorn algorithm, but the improvement in the numerical stability is significant.

Nevertheless, in practice, the extreme-value issues are still not resolved completely by Algorithm 2 due to the exponentiation of the kernel matrix $\boldsymbol{P}^i$. Moreover, we only check for numerical issues once per iteration for efficiency. Note that multiple numerical issue checks can be done in an iteration as a trade-off between computational overhead and stability. Hence, tuning for the cost matrix $\boldsymbol{C}$ magnitudes and $\lambda$, for the values inside the $\exp$ function to not become too extreme, is still required for numerical stability.

## F   Uniform And Regular Polytopes

We provide a brief discussion on the $d$-dimensional uniform and regular polytope families used in the paper (cf. Section 3). For a comprehensive introduction, we refer to [29, 54]. In geometry, regular polytopes are the generalization in any dimensions of regular polygons (e.g., square, hexagon) and regular polyhedra (e.g., simplex, cube). The regular polytopes have their elements as $j$-facets ($0 \leq j \leq d$) - also called cells, faces, edges, and vertices - being transitive and also regular sub-polytopes of dimension $\leq d$ [29]. Specifically, the polytope's facets are pairwise congruent: there exists an isometry that maps any facet to any other facet.

To compactly identify regular polytopes, a *Schläfli symbol* is defined as the form $\{a, b, c, ..., y, z\}$, with regular facets as $\{a, b, c, ..., y\}$, and regular vertex figures as $\{b, c, ..., y, z\}$. For example,

- a polygon having $n$ edges is denoted as $\{n\}$ (e.g., a square is denoted as $\{4\}$),
- a regular polyhedron having $\{n\}$ faces with $p$ faces joining around a vertex is denoted as $\{n, p\}$ (e.g., a cube is denoted as $\{4, 3\}$) and $\{p\}$ is its *vertex figure* (i.e., a figure of an exposed polytope when one vertex is "sliced off"),
- a regular 4-polytope having cells $\{n, p\}$ with $q$ cells joining around an edge is denoted as $\{n, p, q\}$ having vertex figure $\{p, q\}$, and so on.

A *d-dimensional uniform polytope* is a generalization of a regular polytope - only retaining the vertex-transitiveness (i.e., only vertices are pairwise congruent), and is bounded by its uniform facets. In fact, nearly every uniform polytope can be constructed by Wythoff constructions, such as *rectification*, *truncation*, and *alternation* from either regular polytopes or other uniform polytopes [54]. This implies a vast number of possible choices of vertex-transitive uniform polytopes that can be applied to the Sinkhorn Step. Further research in this direction is interesting.

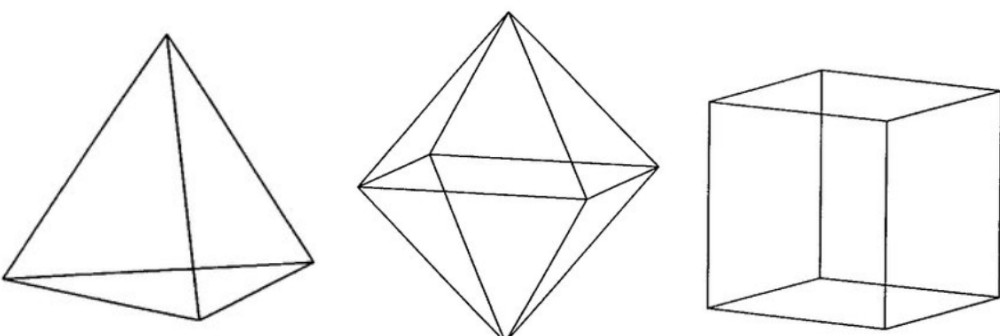

Figure 5: Examples of (left to right) 3-simplex, 3-orthorplex, 3-cube.

Table 3: Regular and uniform polytope families.

| Dimension | Simplices | Orthoplexes | Hypercubes |
|---|---|---|---|
| $d = 2$ | regular trigon $\{3\}$ | square $\{4\}$ | square $\{4\}$ |
| $d = 3$ | regular tetrahedron $\{3, 3\}$ | regular octahedron $\{3, 4\}$ | cube $\{4, 3\}$ |
| Any $d$ | $d$-simplex $\{3^{d-1}\}$ | $d$-orthoplex $\{3^{d-2}, 4\}$ | $d$-cube $\{4^{d-2}, 3\}$ |

We present three families of regular and uniform polytopes in Table 3, which are used in this work due to their construction simplicity (see Fig. 5), and their existence for any dimension. Note that there are regular and uniform polytope families that do not exist in any dimension [29]. The number of vertices is $n = d + 1$ for a $d$-simplex, $n = 2d$ for a $d$-orthoplex, and $n = 2^d$ for a $d$-cube.

**Construction**. We briefly discuss the vertex coordinate construction of $d$-regular polytopes $\mathcal{P}$ inscribing a $(d - 1)$-unit hypersphere with its centroid at the origin. Note that these constructions are GPU vectorizable. First, we denote the standard basis vectors $\boldsymbol{e}_1, \ldots, \boldsymbol{e}_d$ for $\mathbb{R}^d$.

For a regular $d$-simplex, we begin the construction with the *standard* $(d - 1)$-simplex, which is the convex hull of the standard basis vectors $\Delta^{d-1} = \{\sum_{i=1}^d w_i \boldsymbol{e}_i \in \mathbb{R}^d \mid \sum_{i=1}^d w_i = 1, w_i > 0, \text{for } i = 1, \ldots, d\}$. Now, we already got $d$ vertices with the pairwise distance of $\sqrt{2}$. Next, the final vertex lies on the line perpendicular to the barycenter of the standard simplex, so it has the form $(a/d, \ldots, a/d) \in \mathbb{R}^d$ for some scalar $a$. For the final vertex to form regular $d$-simplex, its distances to any other vertices have to be $\sqrt{2}$. Hence, we arrive at two choices of the final vertex coordinate $\frac{1}{d}(1 \pm \sqrt{1 + d})\boldsymbol{1}_d$. Finally, we shift the regular $d$-simplex centroid to zero and rescale the coordinate such that its circumradius is 1, resulting in two sets of $d + 1$ coordinates

$$\left( \sqrt{1 + \frac{1}{d}} \boldsymbol{e}_i - \frac{1}{d\sqrt{d}}(1 \pm \sqrt{d+1})\boldsymbol{1}_d \right) \text{ for } 1 \leq i \leq d, \text{ and } \frac{1}{\sqrt{d}}\boldsymbol{1}_d. \tag{30}$$

Note that we either choose two coordinate sets by choosing $+$ or $-$ in the computation.

For a regular $d$-orthoplex, the construction is trivial. The vertex coordinates are the positive-negative pair of the standard basis vectors, resulting in $2d$ coordinates

$$\boldsymbol{e}_1, -\boldsymbol{e}_1, \ldots, \boldsymbol{e}_d, -\boldsymbol{e}_d \tag{31}$$

For a regular $d$-cube, the construction is also trivial. The vertex coordinates are constructed by choosing each entry of the coordinate $1/2$ or $-1/2$, resulting in $2^d$ vertex coordinates.

# G  d-Dimensional Random Rotation Operator

We describe the random $d$-dimensional rotation operator applied on polytopes mentioned in Section 4. Focusing on the computational perspective, we describe the rotation in any dimension through the lens of matrix eigenvalues. For any $d$-dimensional rotation, a (proper) rotation matrix $\boldsymbol{R} \in \mathbb{R}^{d \times d}$ acting on $\mathbb{R}^d$ is an orthogonal matrix $\boldsymbol{R}^\mathsf{T} = \boldsymbol{R}^{-1}$, leading to $\det(\boldsymbol{R}) = 1$. Roughly speaking, $\boldsymbol{R}$ does not apply contraction or expansion to the polytope convex hull $\text{vol}(\boldsymbol{D}^P) = \text{vol}(\boldsymbol{D}^P \boldsymbol{R})$.

For even dimension $d = 2m$, there exist $d$ eigenvalues having unit magnitudes $\varphi = e^{\pm i\theta_l}$, $l = 1, \ldots, m$. There is no dedicated fixed eigenvalue $\varphi = 1$ depicting the axis of rotation, and thus no axis of rotation exists for even-dimensional spaces. For odd dimensions $d = 2m + 1$, there exists at least one fixed eigenvalue $\varphi = 1$, and the axis of rotation is an odd-dimensional subspace. To see this, set $\varphi = 1$ in $\det(\boldsymbol{R} - \varphi \boldsymbol{I})$ as follows

$$\begin{aligned} \det(\boldsymbol{R} - \boldsymbol{I}) &= \det(\boldsymbol{R}^\mathsf{T})\det(\boldsymbol{R} - \boldsymbol{I}) = \det(\boldsymbol{R}^\mathsf{T}\boldsymbol{R} - \boldsymbol{R}^\mathsf{T}) \\ &= \det(\boldsymbol{I} - \boldsymbol{R}) = (-1)^d \det(\boldsymbol{R} - \boldsymbol{I}) = -\det(\boldsymbol{R} - \boldsymbol{I}), \end{aligned} \tag{32}$$

with $(-1)^d = -1$ for odd dimensions. Hence, $\det(\boldsymbol{R} - \boldsymbol{I}) = 0$. This implies that the corresponding eigenvector $\boldsymbol{r}$ of $\varphi = 1$ is a fixed axis of rotation $\boldsymbol{R}\boldsymbol{r} = \boldsymbol{r}$. When there are some null rotations in the even-dimensional subspace orthogonal to $\boldsymbol{r}$, i.e., when fixing some $\theta_l = 0$, an even number of real unit eigenvalues appears, and thus the total dimension of rotation axis is odd. In general, the odd-dimensional $d = 2m + 1$ rotation is parameterized by the same number $m$ of rotation

angles as in the $2m$-dimensional rotation. As a remark, in $d \geq 4$, there exist pairwise orthogonal planes of rotations, each parameterized by a rotation angle $\theta$. Interestingly, if we smoothly rotate a 4-dimensional object from a starting orientation and choose rotation angle rates such that $\theta_1 = w\theta_2$ with $w$ is an irrational number, the object will never return to its starting orientation.

**Construction.** We only present random rotation operator constructions that are straightforward to vectorize. More methods on any dimensional rotation construction are presented in [55]. For an even-dimensional space $d = 2m$, by observing the complex conjugate eigenvalue pairs, the rotation matrix can be constructed as a block diagonal of $2 \times 2$ matrices

$$\boldsymbol{R}_l = \begin{bmatrix} \cos(\theta_l) & -\sin(\theta_l) \\ \sin(\theta_l) & \cos(\theta_l) \end{bmatrix}, \tag{33}$$

describing a rotation associated with the rotation angle $\theta_l$ and the pairs of eigenvalues $e^{\pm i\theta_l}$, $l = 1, \ldots, m$. In fact, this construction constitutes a *maximal torus* in the special orthogonal group $SO(2m)$ represented as $T(m) = \{\text{diag}(e^{i\theta_1}, \ldots, e^{i\theta_m}), \; \forall l, \theta_l \in \mathbb{R}\}$, describing the set of all simultaneous component rotations in any fixed choice of $m$ pairwise orthogonal rotation planes [56]. This is also a maximal torus for odd-dimensional rotations $SO(2m + 1)$, where the group action fixes the remaining direction. For instance, the maximal tori in $SO(3)$ are given by rotations about a fixed axis of rotation, parameterized by a single rotation angle. Hence, we construct a random $d \times d$ rotation matrix by first uniformly sampling the angle vector $\boldsymbol{\theta} \in [0, 2\pi]^m$, then computing in batch the $2 \times 2$ matrices Eq. (33), and finally arranging them as block diagonal matrix.

Fortunately, in this paper, planning in first-order trajectories always results in an even-dimensional state space. Hence, we do not need to specify the axis of rotation. For general construction of a uniformly random rotation matrix in any dimension $d \geq 2$, readers can refer to the Stewart method [57] and our implementation of Steward method at `https://github.com/anindex/ssax/blob/main/ssax/ss/rotation.py#L38`.

# H   Related Works

**Motion optimization.** While sampling-based motion planning algorithms have gained significant traction [5, 6], they are typically computationally expensive, hindering their application in real-world problems. Moreover, these methods cannot guarantee smoothness in the trajectory execution, resulting in jerky robot motions that must be post-processed before executing them on a robot [11]. To address the need for smooth trajectories, a family of gradient-based methods was proposed [7, 21, 8] for finding locally optimal solutions. These methods require differentiable cost functions, effectively requiring crafting or learning signed-distance fields of obstacles. CHOMP [7] and its variants [58, 59, 50] optimize a cost function using covariant gradient descent over an initially suboptimal trajectory that connects the start and goal configuration. However, such approaches can easily get trapped in local minima, usually due to bad initializations. Stochastic trajectory optimizers, e.g., STOMP [9] sample candidate trajectories from proposal distributions, evaluate their cost, and weigh them for performing updates [52, 13]. Although gradient-free methods can handle discontinuous costs (e.g., planning with surface contact), they may cause oscillatory behavior or failure to converge, requiring additional heuristics for acquiring better performance [60]. Schulman et al. [61] addresses the computational complexity of CHOMP and STOMP, which require fine trajectory discretization for collision checking, proposing a sequential quadratic program with continuous time collision checking. Gaussian Process Motion Planning (GPMP) [8] casts motion optimization as a probabilistic inference problem. A trajectory is parameterized as a function of continuous-time that maps to robot states, while a GP is used as a prior distribution to encourage trajectory smoothness, and a likelihood function encodes feasibility. The trajectory is inferred via maximum a Posteriori (MAP) estimation from the posterior distribution of trajectories, constructed out of the GP prior and the likelihood function. In this work, we perform updates on waypoints across multiple trajectories concurrently. This view is also considered in methods that resolve trajectory optimization via collocation [62].

**Optimal transport in robot planning.** While OT has several practical applications in problems of resource assignment and machine learning [17], its application to robotics is scarce. Most applications consider swarm and multi-robot coordination [63–67], while OT can be used for exploration during planning [68] and for curriculum learning [69]. A comprehensive review of OT in control is available in [70]. Recently, Le et al. [71] proposed a method for re-weighting Riemannian motion policies [72]

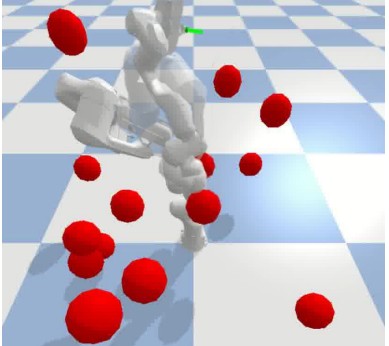 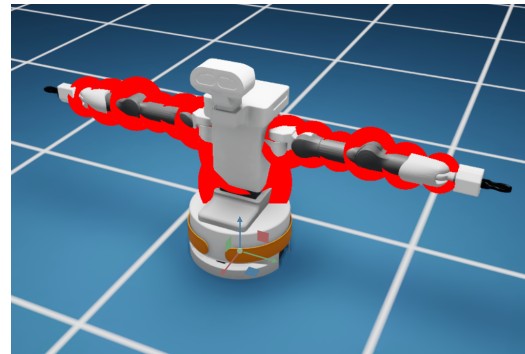

Figure 6: (Left) An example of the Panda arm plan execution for three simulation frames. The green line denotes a $SE(3)$ goal. (Right) An example of red collision spheres attached to TIAGo++ mesh at a configuration. The collision spheres are transformed with the robot links via forward kinematics.

using an unbalanced OT at the high level, leading to fast reactive robot motion generation that effectively escapes local minima.

# I    Additional Experimental Details

We elaborate on all additional experimental details omitted in the main paper. All experiments are executed in a single RTX3080Ti GPU and a single AMD Ryzen 5900X CPU. Note that due to the fact that all codebases are implemented in PyTorch (e.g., forward kinematics, planning objectives, collision checkings, environments, etc.), hence due to conformity reasons, we also implement RRT*/I-RRT* in PyTorch. However, we set using CPU when running RRT*/I-RRT* experiments and set using GPU for MPOT and the other baselines. An example of Panda execution for collision checking in PyBullet is shown in Fig. 6.

For a fair comparison, we construct the initialization GP prior $\mathcal{N}(\boldsymbol{\mu}_0, \boldsymbol{K}_0)$ with a constant-velocity straight line connecting the start and goal configurations, and sample initial trajectories for all trajectory optimization algorithms. We use the constant-velocity GP prior Eq. (19), both in the cost term and for the initial trajectory samples. To the best of our knowledge, the baselines are not explicitly designed for batch trajectory optimization. Striving for a unifying experiment pipeline and fair comparison, we reimplement all baselines in PyTorch with vectorization design (beside RRT*) and fine-tune them with the parallelization setting, which is unavailable in the original codebases.

Notably, we use RRT*/I-RRT* as a feasibility indicator of the environments since they enjoy probabilistic completeness, i.e., at an infinite time budget if a solution exists these search-based methods will find the plan. Optimization-based motion planners, like MPOT, GPMP2, CHOMP, and STOMP are only local optimizers. Therefore, if a solution cannot be found by RRT*/I-RRT*, then it is not possible that optimization-based approaches can recover a solution.

## I.1    MPOT experiment settings

For MPOT, we apply $\epsilon$-annealing, normalize the configuration space limits (e.g., position limits, joint limits) into the $[-1, 1]$ range, and do the Sinkhorn Step in the normalized space. MPOT is cost-sensitive due to exponential terms inside the Sinkhorn algorithm, hence, in practice, we normalize the cost matrix to the range $[0, 1]$. The MPOT hyperparameters used in the experiments are presented in Table 4.

## I.2    Environments

For the *point-mass* environment, we populate 15 square and circle obstacles randomly and uniformly inside x-y limits of $[-10, 10]$, with each obstacle having a radius or width of 2 (cf. Fig. 1). We generate 100 environment-seeds, and for each environment-seed, we randomly sample 10 collision-free pairs of start and goal states, resulting in 1000 planning tasks. We plan each task in parallel 100 trajectories of horizon 64. A trajectory is considered successful if collision-free.

For the *Panda* environment, we also generate 100 environment-seeds. Each environment-seed contains randomly sampled 15 obstacle-spheres having a radius of 10cm inside the x-y-z limits of

Table 4: Experiment hyperparameters of MPOT. $\alpha_0, \beta_0$ are the initial stepsize and probe radius. $h$ is the number of probe points per search direction. $eps$ is the annealing rate. $P$ is the polytope type, and $\lambda$ is the entropic scaling of OT problem.

|  | Point-mass | Panda | TIAGo++ |
|---|---|---|---|
| $\alpha_0$ | 0.38 | 0.03 | 0.03 |
| $\beta_0$ | 0.5 | 0.15 | 0.1 |
| $h$ | 10 | 3 | 3 |
| $\epsilon$ | 0.032 | 0.035 | 0.05 |
| $P$ | $d$-cube | $d$-orthoplex | $d$-orthoplex |
| $\lambda$ | 0.01 | 0.01 | 0.01 |

$[[-0.7, 0.7], [-0.7, 0.7], [0.1, 1.]]$ (cf. Fig. 6), ensuring that the Panda's initial configuration has no collisions. Then, we sample 5 random collision-free (including self-collision-free) configurations, we check with RRT* the feasibility of solutions connecting initial and goal configurations, and then compute the $SE(3)$ pose of the end-effector as a possible goal. Thus, we create a total of 500 planning tasks and plan in parallel 10 trajectories containing 64 timesteps. To construct the GP prior, we first solve inverse kinematics (IK) for the $SE(3)$ goal in PyBullet, and then create a constant-velocity straight line to that goal. A trajectory is considered successful when the robot reaches the $SE(3)$ goal within a distance threshold with no collisions.

In the *TIAGo++* environment, we design a realistic high-dimensional mobile manipulation task in PyBullet (cf. Fig. 4). The task comprises two parts: the fetch part and place part; thus, it requires solving two planning problems. Each plan contains 128 timesteps, and we plan a single trajectory for each planner due to the high computational and memory demands. We generate 20 seeds by randomly spawning the robot in the room, resulting in 20 tasks in total. To sample initial trajectories with the GP, we randomly place the robot's base at the front side of the table or the shelf and solve IK using PyBullet. We designed a holonomic base for this experiment. A successful trajectory finds collision-free plans, successfully grasping the cup and placing it on the shelf.

## I.3 Metrics

Comparing various aspects among different types of motion planners is challenging. We aim to benchmark the capability of planners to parallelize trajectory optimization under dense environment settings. We tune all baselines to the best performance possible for the respective experimental settings and then set the convergence threshold and a maximum number of iterations for each planner.

In all experiments, we consider $N_s$ environment-seeds and $N_t$ tasks for each environment-seed. For each task, we optimize $N_p$ plans having $T$ horizon.

**Planning Time**. We aim to benchmark not only the success rate but also the *parallelization quality* of planners. Hence, we tune all baselines for each experiment, and then measure the planning time T$[s]$ of trajectory optimizers until convergence or till maximum iteration is reached. T$[s]$ is averaged over $N_s \times N_t$ tasks.

**Success Rate**. We measure the success rate of task executions over environment-seeds. Specifically, SUC$[\%] = N_{st}/N_t \times 100$ with $N_{st}$ being the number of successful task executions (i.e., having at least a successful trajectory in a batch). The success rate is averaged over $N_s$ environment-seeds.

**Parallelization Quality**. We measure the parallelization quality, reflecting the success rate of trajectories in a single task. Specifically, GOOD$[\%] = N_{sp}/N_p \times 100$ with $N_{sp}$ being the number of successful trajectories in a task, and it is averaged over $N_s \times N_t$ tasks.

**Smoothness**. We measure changing magnitudes of the optimized velocities as smoothness criteria, reflecting energy efficiency. This measure can be interpreted as accelerations multiplied by the time discretization. Specifically, S $= \frac{1}{T} \sum_{t=0}^{T-1} \|\dot{\mathbf{x}}_{t+1} - \dot{\mathbf{x}}_t\|$. S is averaged over successful trajectories in $N_s \times N_t$ tasks.

**Path Length**. We measure the trajectory length, reflecting natural execution and also smoothness. Specifically, PL $= \sum_{t=0}^{T-1} \|\mathbf{x}_{t+1} - \mathbf{x}_t\|$. PL is averaged over successful trajectories in $N_s \times N_t$ tasks.

## I.4 Motion planning costs

For the obstacle costs, we use an occupancy map having binary values for gradient-free planners (including MPOT) while we implement signed distance fields (SDFs) of obstacles for the gradient-based planners. For self-collision costs, we use the common practice of populating with spheres the robot mesh and transforming them with forward kinematics onto the task space [7, 8]. To be consistent for all planners, joint limits are enforced as an L2 cost for joint violations. Besides the point-mass experiment, all collisions are checked by PyBullet. The differentiable forward kinematics implemented in PyTorch is used for all planners.

**Goal Costs**. For the $SE(3)$ goal cost, given two points $\boldsymbol{T}_1 = [\boldsymbol{R}_1, \boldsymbol{p}_1]$ and $\boldsymbol{T}_2 = [\boldsymbol{R}_2, \boldsymbol{p}_2]$ in $SE(3)$, we decompose a translational and rotational part, and choose the following distance as cost $d_{SE(3)}(\boldsymbol{T}_1, \boldsymbol{T}_2) = \|\boldsymbol{p}_1 - \boldsymbol{p}_2\| + \|(\text{LogMap}(\boldsymbol{R}_1^\mathsf{T} \boldsymbol{R}_2))\|$, where $\text{LogMap}(\cdot)$ is the operator that maps an element of $SO(3)$ to its tangent space [73].

**Collision Costs**. Similar to CHOMP and GPMP2, we populate $K$ *collision spheres* on the robot body (shown in Fig. 6). Given differentiable forward kinematics implemented in PyTorch (for propagating gradients back to configuration space), the obstacle cost for any configuration $\boldsymbol{q}$ is

$$C_{\text{obs}}(\boldsymbol{q}) = \frac{1}{K} \sum_{j=1}^{K} c(\boldsymbol{x}(\boldsymbol{q}, S_j)) \tag{34}$$

with $\boldsymbol{x}(\boldsymbol{q}, S_j)$ is the forward kinematics position of the $j^{\text{th}}$-collision sphere, which is computed in batch. For gradient-based motion optimizers, we design the cost using the signed-distance function $d(\cdot)$ from the sphere center to the closest obstacle surface (plus the sphere radius) in the task space with a $\epsilon > 0$ margin

$$c(\boldsymbol{x}) = \begin{cases} d(\boldsymbol{x}) + \epsilon & \text{if } d(\boldsymbol{x}) \geq -\epsilon \\ 0 & \text{if } d(\boldsymbol{x}) < -\epsilon \end{cases}. \tag{35}$$

For gradient-free planners, we discretize the collision spheres into fixed probe points, check them in batch with the occupancy map, and then average the obstacle cost over probe points.

**Self-collision Costs**. We group the collision spheres that belong to the same robot links. Then, we compute the pair-wise link sphere distances. The self-collision cost is the average of the computed pair-wise distances.

**Joint Limits Cost**. We also construct a soft constraint on joint limits (and velocity limits) by computing the L2 norm violation as cost, with a $\epsilon > 0$ margin on each dimension $i$

$$C_{\text{limits}}(q_i) = \begin{cases} \|q_{\min} + \epsilon - q_i\| & \text{if } q_i < q_{\min} + \epsilon \\ 0 & \text{if } q_{\min} + \epsilon \leq q_i \leq q_{\max} - \epsilon \\ \|q_{\max} - \epsilon - q_i\| & \text{else} \end{cases}. \tag{36}$$

# J Ablation Study

In this section, we study different algorithmic aspects, horizons and number of paralleled plans, and also provide an ablation on polytope choices.

## J.1 Algorithmic ablations

We study the empirical convergence and the parallelization quality over Sinkhorn Steps between the main algorithm MPOT and its variants: MPOT-NoRot - no random rotation applied on the polytopes, and MPOT-NoAnnealing - annealing option is disabled. This ablation study is conducted on the point-mass experiment due to the extremely narrow passages and non-smooth, non-convex objective function, contrasting the performance difference between algorithmic options.

The performance gap between MPOT-NoRot and the others in Fig. 7 is significant. The absence of random rotation on waypoint polytopes leads to biases in the planning cost approximation due to the fixed *probe set* $H^P$. This approximation bias from non-random rotation becomes more prominent in higher-dimensional tasks due to the sparse search direction set. This experiment result confirms the robustness gained from the random rotation for arbitrary objective function conditions.

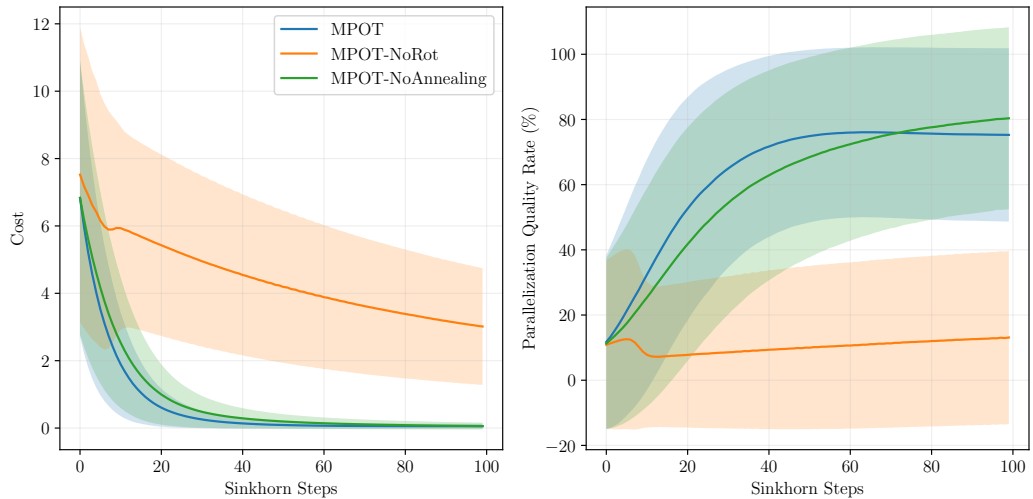

Figure 7: Ablation study on algorithmic choices in the point-mass environment. All planners are terminated at 100 Sinkhorn Steps. All statistics are evaluated on 1000 tasks as described in Section 5.2.

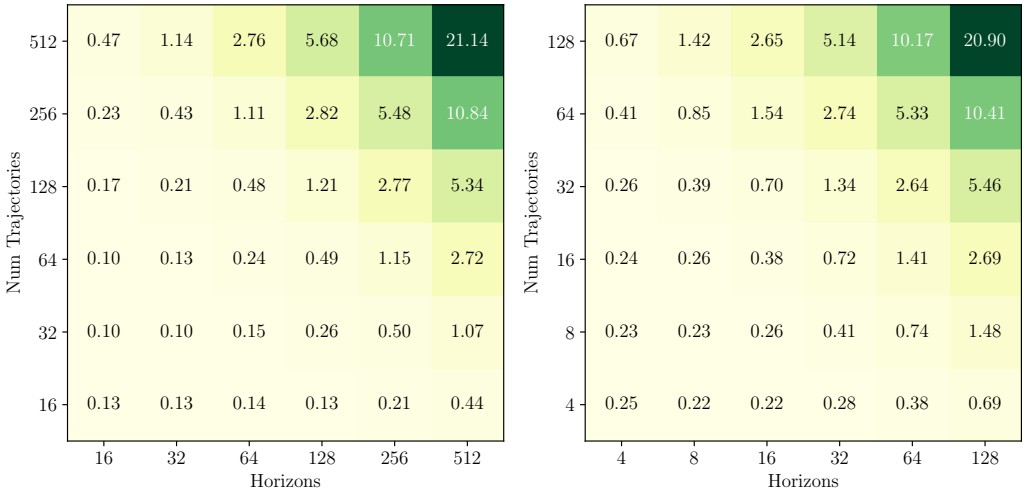

Figure 8: Planning time heatmap in seconds while varying the horizons and number of paralleled trajectories on both the point-mass (left) and the Panda (right) environments.

Between MPOT and MPOT-NoAnnealing, the performance gap depends on the context. MPOT has a faster convergence rate due to annealing the step and probe radius, which leads to a better approximation of local minima. However, it requires careful tuning of the annealing rate to avoid premature convergence and missing better local minima. MPOT-NoAnnealing converges slower and thus takes more time, but eventually discovers more successful local minima (nearly 80%) than MPOT with annealing (cf. Table 1). This is a trade-off between planning efficiency and parallelization quality with the annealing option.

## J.2 Flattening ablations

In both the point-mass and the Panda environments, we experiment with different horizons $T$ and the number of parallel plans $N_p$. For each $(T, N_p)$ combination, we tune MPOT to achieve a satisfactory success rate and then measure the planning time until convergence, as shown in Fig. 8. The planning time heatmap highlights the batch computation property of MPOT, resulting in a nearly symmetric pattern. Despite long horizons and large batch trajectories, the planning time remains reasonable

Table 5: Polytope ablation study on the Panda environment. All statistics are evaluated on 500 tasks as described in Section 5.2.

| | T[s] | SUC[%] | GOOD[%] | S | PL |
|---|---|---|---|---|---|
| MPOT-Random | $2.5 \pm 0.0$ | $70.1 \pm 23.7$ | $58.3 \pm 44.3$ | $0.03 \pm 0.01$ | $4.7 \pm 1.2$ |
| MPOT-Simplex | $\mathbf{0.5} \pm 0.0$ | $65.8 \pm 24.5$ | $52.1 \pm 45.3$ | $0.01 \pm 0.01$ | $4.6 \pm 1.1$ |
| MPOT-Orthoplex | $0.8 \pm 0.1$ | $\mathbf{71.6} \pm 23.2$ | $\mathbf{60.2} \pm 44.4$ | $0.01 \pm 0.01$ | $4.6 \pm 0.9$ |

(under a minute) and can be run efficiently on a single GPU without excessive memory usage, making it suitable, for example, for collecting datasets for learning neural network models.

## J.3 Polytope ablations

In Table 5, we compare the performance of MPOT-Orthoplex (i.e., MPOT in the Panda experiments) with its variants: MPOT-Simplex using the $d$-simplex vertices as search direction set $D^P$, and MPOT-Random, i.e., not using any polytope structure. For MPOT-Random, we generate 100 points on the 13-sphere ($d = 14$ for the Panda environment) as the search direction set $D$ for each waypoint at each Sinkhorn Step, using the Marsaglia method [74]. As expected, since the $d$-simplex has fewer vertices than the $d$-orthoplex, MPOT-Simplex has better planning time but sacrifices some success rate due to a more sparse approximation. MPOT-Random, while achieving a comparable success rate, performs even worse in both planning time and smoothness criteria. We also observe that increasing the number of sampled points on the sphere improves the smoothness marginally. However, increasing the sample points worsens the planning time in general, inducing more matrix columns and instabilities in the already large dimension cost matrix (cf. Appendix E) of the OT problem. This ablation study highlights the significance of the polytope structure for the Sinkhorn Step in high-dimensional settings.

## J.4 Smooth gradient approximation ablations

We conduct an ablation on the gradient approximation of Sinkhorn Step w.r.t. different important hyperparameter settings for sanity check of Sinkhorn Step's optimization behavior on a smooth objective function. We choose the Styblinski-Tang function (cf. Fig. 9) in 10D as the smooth objective function due to its variable dimension and multi-modality for non-convex optimization benchmark [75]. We target the most important hyperparameters of *polytope type $P$*, and entropic regularization scalar $\lambda$. These parameters sensitively affect the Sinkhorn Step's optimization performance. We set the other important hyperparameters of *step size* and *probe size* $\alpha = \beta = 0.1$ to be constant, the number of probing points per vertices to be 5 and turn off the annealing option for all optimization runs. The cosine similarity is defined for each particle $\boldsymbol{x}_i \in X$ as follows:

$$\text{CS}_i = \frac{\boldsymbol{s}(\boldsymbol{x}_i) \cdot (-\nabla f(\boldsymbol{x}_i))}{\|\boldsymbol{s}(\boldsymbol{x}_i)\| \, \|-\nabla f(\boldsymbol{x}_i)\|} \tag{37}$$

Regarding this smooth objective, we observe the gradient approximation quality is consistent with Lemma 1, with increasing cosine similarities for all curves from left to right column (cf. Fig. 10). However, regarding entropic regularization scalar $\lambda$, we observe higher cosine similarity and lower curve variance for larger $\lambda$. Interestingly, this means higher $\lambda$ induces both computational benefit solving entropic OT [19] and higher *entropic smoothing bias* [76], where the latter regularizes the gradient approximation directions, while it contrarily blurs the result barycenters in the barycenter problem. Notably, this Sinkhorn Step smoothing effect is more necessary in the case of $P = $ cube toward the end of optimization (i.e., near the local minima/fixed points), where the gradients have small magnitudes and may be noisy while the Sinkhorn Step size is constant. High $\lambda = 0.5$ (red curve) keeps high cosine similarity toward fixed points (cf. Fig. 10), while the lower/sharper $\lambda$ exhibits degradation due to noisy random rotated 10-cube with constant-size having $2^{10}$ vertices.

Note that the conclusion drawn from this ablation may not apply to the motion planning application in the main paper since we are evaluating the Sinkhorn Step on smooth objective functions, while the motion planning costs may have an ill-formed cost landscape. Further investigation of (sub)-gradient approximation in various objective function conditions is very interesting for future work.

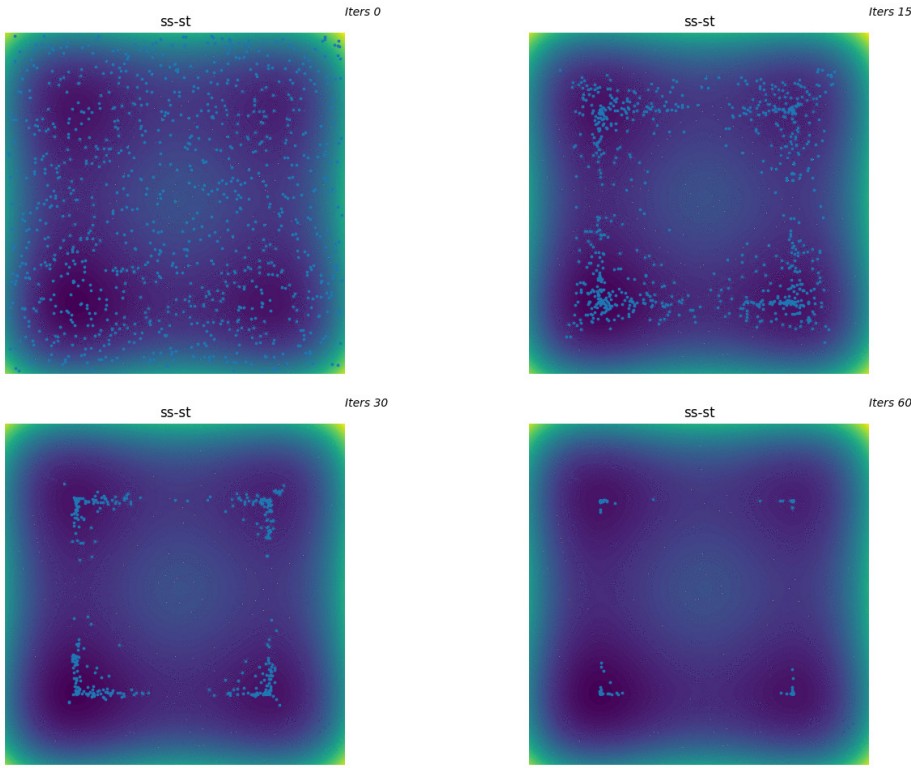

Figure 9: An example optimization run of 1000 points on the Styblinski-Tang function with Sinkhorn Step. The points are uniformly sampled at the start of optimization. This plot shows the projected optimization run in the first two dimensions.

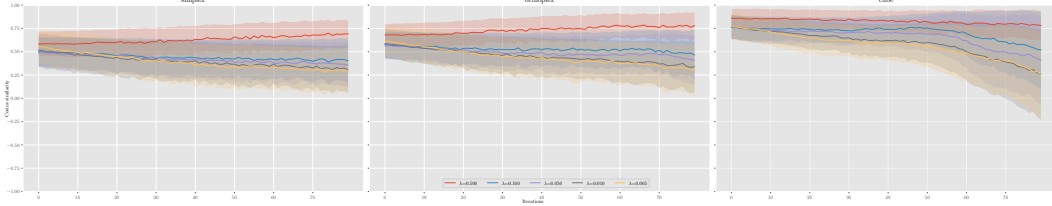

Figure 10: Ablation study on gradient approximation with cosine similarity between Sinkhorn Step directions and true gradients. We choose the Styblinski-Tang function as the test objective function. Each curve represents an optimization run of 1000 points w.r.t to entropic regularization scalar $\lambda$ and polytope choice (corresponding to each column), where each iteration shows the mean and variance of cosine similarity of points w.r.t their true gradients. We conduct 50 seeds for each curve, where for all seeds we concatenate the cosine similarities of all optimizing points across the seeds at each iteration.

