# OpenReview forum: "Accelerating Motion Planning via Optimal Transport"
_NeurIPS.cc/2023/Conference — NeurIPS 2023 poster_

### Official Review · Reviewer_hcju · 2023-06-20

**Soundness:** 3 good
**Presentation:** 3 good
**Contribution:** 2 fair
**Rating:** 6
**Confidence:** 3

**Summary:**

With the aim of improving efficiency in motion planning, this paper proposes an efficient, gradient-free optimization method, MPOT. This is enabled by the introducing the Sinkhorn step, a zero-order parallelizable update rule that is guaranteed to converge under smoothness and boundedness assumptions. The authors perform empirical studies of the effectiveness of their method in 3 benchmarks.

**Strengths:**

- Motion planning is an important task that must be solved quickly while outputting smooth plans to enable the deployment of robotic systems in the real world. As such, this problem is a relevant one.
- Both the Sinkhorn step as MPOT are described in detail and, to the best of my knowledge, are novel contributions to the field.
- The paper is well written, well organized and clear.

**Weaknesses:**

- The benchmarking results presented in Section 5.2 correspond to two settings in cluttered environments where RRT*/I-RRT* do very well despite the running time cost incurred. Including benchmarks where RRT* fails would be interesting to explore further other potential advantages of this method besides run time complexity.
- In Section 5.3, the authors only report results on MPOT and GPMP2, despite the fact that methods like SGPMP solve similar-sized problems in their experiments. A discussion on the limitations of application, or extension of the benchmark would make the case stronger within this setting.

**Minor comments**:
- In line 386, it should read “At last”.

**Questions:**

- On what hardware is the time comparison from Table 1 being drawn? RRT*/I-RRT* will benefit from stronger CPUs, whereas the other methods have GPU dependencies, so a hardware comparison is important to understand the run time benchmark.
- How does MPOT do in other benchmarks where RRT* fails to reach a solution?
- Why is MPOT only compared to GPMP2 in the mobile manipulation experiment, instead of considering similar baselines to the previous cases?
- Are there other known applications for the Sinkhorn step? The authors comment “sampling methods or variational inference”, but a brief slightly more detailed discussion of this in the paper could improve the contributions greatly.

**Limitations:**

The authors have addressed limitations of their work to a good extent in Section 5.

---

> ### Author Rebuttal · Authors · 2023-08-08
>
> Thank you very much for your insightful comments and constructive feedback.
>
> **Regarding the comment about RRT\*/I-RRT\*:** Initially, our intention was to use RRT*/I-RRT* as an indicator of feasibility of the tested environments since they enjoy probabilistic completeness, i.e., at infinite time budget if a solution exists these search-based methods will find the plan. Optimization-based motion planners, like MPOT, GPMP2, CHOMP, and STOMP are only local optimizers. Therefore, if a solution cannot be found by RRT*/I-RRT*, then it is not possible optimization-based approaches can recover a solution. Please note that key issues with RRT*/I-RRT* are the computational complexity and the lack of smoothness (e.g., see the answer for question 2, about the performance of RRT*/I-RRT* for the TIAGo++ environment). Utilizing a motion planning strategy like MPOT that plans very high frequencies, we can potentially optimize in real-time and be more efficient in re-planning in dynamic environments, while the induced smoothness of MPOT makes the trajectory tracking from the low-level controller much easier. We plan to investigate reactive motion planning with MPOT in future work.
>
> 1. **What hardware is the time comparison from Table 1 being drawn?** All experiments are executed in a single RTX3080Ti GPU and a single AMD Ryzen 5900X CPU. Note that due to the fact that all codebases are implemented in PyTorch (e.g., forward kinematics, planning objectives, collision checkings, environments, etc.), hence due to conformity reasons, we also implement RRT*/I-RRT* in PyTorch. However, we set using CPU  when running RRT*/I-RRT* experiments and set using GPU for MPOT and other baselines.
>
> 2. **How does MPOT do in other benchmarks where RRT\* fails to reach a solution?** In our experiments, RRT*/I-RRT* only fails to find solutions in TIAGo++ environment due to hitting the time-limit budget. We tried a very high time limit (1000 seconds); however, the dimensionality of TIAGo++ environment is too high for RRT* to explore to reach the narrow grasp point. MPOT manages to find a reasonable solution in a few seconds in this case due to being a batch gradient-free optimizer leveraging the effective Sinkhorn algorithm.
>
> 3. **Why is MPOT only compared to GPMP2 in the mobile manipulation experiment, instead of considering similar baselines to the previous cases?** We conducted additional comparative experiments on the TIAGo++ environment to better support our claims, and according to the reviewer's request, the results are available in the rebuttal attachment. Apparently, CHOMP performs worse than GPMP2 and takes more iterations in a cluttered environment. However, according to the new table result, CHOMP beats GPMP2 in run time complexity in TIAGo++ environment due to its simpler update rule. Following your suggestion, we tried SGPMP and observed that we could not tune SGPMP to surpass the success rate 5/20; hence we opted for the minimum iterations to achieve the mentioned success rate. The problem lies in how SGPMP explores with constant GP variance in a high dimensional setting; thus, a possible future extension of SGPMP with adaptive explorative variance could mitigate the problem. Similarly, STOMP's exploration mechanism is even more restrictive, and we could not tune it to work in this environment. Overall, with the efficient Sinkhorn Step facilitating individual waypoint exploration, MPOT outperforms the baselines in terms of run time complexity by a considerable margin in this case while maintaining reasonable smoothness and task performance.
>
> 4. **Are there other known applications for the Sinkhorn step?** To the best of our knowledge, we are the first to propose Sinkhorn Step - an optimization operator that utilizes the Sinkhorn algorithm, that connects zero-order optimization in non-convex objectives with Optimal Transport theories. We briefly mention our vision to apply Sinkhorn Step in other Machine Learning fields. For example, Sinkhorn Step can approximate gradients in SVGD [1] with a suitable choice of the polytope family. Another example in variational inference is to apply Sinkhorn Step to update a Gaussian Mixture Model as the proposal distribution to match the unormalized target distribution.
>
> [1] Liu, Qiang, and Dilin Wang. "Stein variational gradient descent: A general purpose bayesian inference algorithm." Advances in neural information processing systems 29 (2016).

---

> > ### Comment · Reviewer_hcju · 2023-08-14
> > **Response to Rebuttal**
> >
> > Thank you for the detailed rebuttal, the authors have answered all of my questions. Adding the extra mobile manipulation results will definitely make the experimental section stronger.

---

### Official Review · Reviewer_2eXt · 2023-07-06

**Soundness:** 3 good
**Presentation:** 3 good
**Contribution:** 3 good
**Rating:** 7
**Confidence:** 3

**Summary:**

This paper proposes MPOT, a gradient-free method that optimizes a batch of smooth trajectories with nonlinear costs even for high dimensional tasks. In especial, a zero-order and highly-parallelizable update rule called Sinkhorn Step is proposed to facilitate the optimization process. MPOT outperforms the baseline methods with respect to the planning speed and success rate, across various tasks.

**Strengths:**

1. This paper addresses a fundamental problem - trajectory smoothing in motion planning, and proposes a solid method.
2. The paper is self-contained and provide very detailed introduction of the method. It is well organized and easy to read.

**Weaknesses:**

1. To enable the batch optimization, it seems that each sequence of optimizing points in the same batch are asked to share the same length, thereby forming a matrix $X \in R^{n\times d}$. Such operation kind of hinders the flexibility of the algorithm to process trajectories with quite different number of waypoints in the batch-wise manner.

**Questions:**

1. Please refer to the weakness. How do you process the sequences with different waypoint numbers?
2. How do you evaluate the planning time of other baselines, such as STOMP and GPMP2? Are other baseline methods also executed in batch?
3. If other baselines are not performed in batch, how do you justify the motivation of performing trajectory smooth in batch?

**Limitations:**

Yes, limitations are addressed by the authors. I see no negative societal impact.

---

> ### Author Rebuttal · Authors · 2023-08-08
>
> We thank the reviewer for their positive evaluation of our work.
>
> 1. **How do you process the sequences with different waypoint numbers?** Currently, for vectorizing the update of all waypoints across the batch of trajectories, we flatten the batch and horizon dimensions and apply Sinkhorn Step, then we reshape the tensor to the original shape. Notice that what really glues the waypoints in the same trajectory together after optimization is the log of the Gaussian Process as cost model, which promotes smoothness and model consistency. Given this pretext, in case of a batch of different horizon trajectories, we address this case by setting maximum horizon $T_{\textrm{max}}$ and padding with zeros for those trajectories having $T < T_{\textrm{max}}$. Then, we also set zeros for all rows corresponding to these padded points in the cost matrix $\mathbf{C}^{T_{\textrm{max}} \times n}$. By this way, the padded points are ignored in the barycentric projection. Intuitively, we just need to manipulate cost entries to dictate the behavior of waypoints.
>
> 2. **How do you evaluate the planning time of other baselines, such as STOMP and GPMP2? Are other baseline methods also executed in batch?** We further describe our experiment settings in Appendix I. We tune all baselines for each experiment and then measure the planning time T$[s]$ of baselines until convergence or till maximum iteration is reached. T$[s]$ is averaged over number of environment-seeds and number of tasks.
>     - Striving for a fair comparison, in all cases, besides RRT*/I-RRT*, we reimplemented all baselines in PyTorch and fine-tuned them with the vectorization option. To the best of our knowledge, the baselines are not explicitly designed for plan vectorization. Thus, their associated public codebases are implemented for single-plan querying.
>     - Regarding the vectorization of RRT and its variants, we found some works [1, 2] that focus on parallelizing specific algorithmic components of RRT*, such as graph operations or collision checking in a single planning instance. They resort to special hardware design (e.g., FPGA) or state-space grid discretization, limiting their application to general settings. Thus, vectorizing the whole RRT* algorithm pipeline is non-trivial and still an open question. We opted for serial computation of RRT* to affirm the environments' solvability due to its completeness property, while reflecting the performance gap between GPU-vectorization optimization-based algorithms and serial classical sampling-based algorithms.
>
> 3. **If other baselines are not performed in batch, how do you justify the motivation of performing trajectory smooth in batch?** As explained above, all baseline comparisons (except RRT*/I-RRT*) are performed in batch. On a further note, there are three interplaying factors that contribute to the solution diversity and, hence discovering better modes:
>     - the step radius of Sinkhorn Step
>     - the moderately-high initialization variances
>     - the number of plans in a batch
>
>     -> For using MPOT as an oracle for collecting datasets, these factors contributing to the solution diversity covering various modes, capturing homotopy classes of the tasks and their associated contexts. For direct execution, abundance of solutions vastly increases the probability of finding good local minima, which we can select the best solution according to some criteria, e.g., collision avoidance, smoothness, model consistency, etc. We will add such a discussion in the final version of the paper to better motivate the need of batch-wise trajectory optimization.
>
> [1] Bialkowski, Joshua, Sertac Karaman, and Emilio Frazzoli. "Massively parallelizing the RRT and the RRT." 2011 IEEE/RSJ International Conference on Intelligent Robots and Systems. IEEE, 2011.
>
> [2] Xiao, Size, Neil Bergmann, and Adam Postula. "Parallel RRT* architecture design for motion planning." 2017 27th International Conference on Field Programmable Logic and Applications (FPL). IEEE, 2017.

---

> > ### Comment · Reviewer_2eXt · 2023-08-17
> >
> > Thank you for the authors' response. My comments and questions have been addressed. I would like to keep the rating as 7.;

---

### Official Review · Reviewer_MARJ · 2023-07-07

**Soundness:** 3 good
**Presentation:** 3 good
**Contribution:** 3 good
**Rating:** 7
**Confidence:** 2

**Summary:**

The paper focuses on the optimization of motion planning problem, by introducing a gradient-free method that is parallelizable and smooth. The method first probes the costs at several vertices, then decides the optimization direction by aligning the weight matrix with the cost matrix. Furthermore, to enforce the weight matrix as a joint distribution, the method introduces Sinkhorn Step, which is based on an acceleration technique for optimal transport optimization. Results show that the method has significantly low planning time and a high success rate.

**Strengths:**

1. Though including some technical mathematical concepts, the paper has good writing and is easy to understand.
2. Results show the method has significant empirical performance.
3. Technical tricks and limitations are well discussed, for example, Section 4.2 and Section 5.3.
4. Implementing most baselines in PyTorch seems to be a lot of work. This is also a contribution, since it provides a platform to compare fairly with previous methods.

**Weaknesses:**

1. To me, 7 DoF is still not high-dimensional tasks (TIAGo++ should be if you are not using action primitives). So since the author claim that the approach could work better than the baselines in high-dimensional tasks, I would appreciate a systematic comparison to other baselines in more high-dimensional tasks, probably in addition to TIAGo++.

**Questions:**

1. How is D^P initialized at the beginning of the algorithm? I guess it is not that important since you will rotate it for each step, but I still would love to know.

**Limitations:**

Limitations are adequately addressed.

I do not see any significant potential negative societal impact.

---

> ### Author Rebuttal · Authors · 2023-08-08
>
> We appreciate the positive feedback on our contributions.
>
> Regarding the reviewer's comments and questions:
>
> - **On the dimensionality of trajectory optimization problem**: We do not consider any movement primitives in our experiments. In all cases, we consider optimizing the full-state (the concatenation of position and velocity) trajectories batch-wise. As mentioned in the experiment description (Sec. 5.1), the state dimension (configuration position and velocity) is $d=4$ for the point-mass experiment, $d=14$ for the Panda experiment, and $d=36$ ($3$ dimensions for the base, $1$ for the torso, and $14$ for the two arms, and their velocities) for the mobile manipulation experiment. Note that the typical optimizing variable would be the whole trajectory, leading to variable dimension $T \times d$. In our vectorization setting, it would be batched $b \times T \times d$. Hence, even for the Panda case, typical full-state motion planning with multiple objectives, such as obstacle avoidance, self-collision avoidance, smoothness, goal-reaching, joint limit handling, etc., is inherently challenging due to many local minima and high computational cost due to many objectives. Notably, we design the TIAGo++ mobile manipulation experiment as a stress test, reflecting even more performance differences between MPOT and the baselines.
>
> - **The importance of full-state trajectory optimization**: Optimizing a full-state (position & velocity) trajectory is vital for many robotics tasks, e.g., obstacle avoidance or human-robot collaboration, where smooth motion is desirable. Full-state reference trajectory generally allows for the low-level controller to track more smoothly. In contrast, if the trajectory is position-only, it is typically challenging to tune the gain of the position controller to track the reference trajectory correctly with large jerks or lower jerks but inaccurate tracking.
>
> - **Additional baseline comparisons on TIAGo++ environment**: We conducted additional comparative experiments on the TIAGo++ environment to better support our claims, and according to the reviewer's request, the results are available in the rebuttal attachment. Apparently, CHOMP performs worse than GPMP2 and takes more iterations in a cluttered environment. However, according to the new table result, CHOMP beats GPMP2 in run time complexity in TIAGo++ environment due to its simpler update rule. Regarding SGPMP, we could not tune SGPMP to surpass the success rate of 5/20; hence we opted for the minimum iterations to achieve the mentioned success rate. The problem lies in how SGPMP explores with constant GP variance in a high dimensional setting; thus, a possible future extension of SGPMP with adaptive explorative variance could mitigate the problem. Similarly, STOMP's exploration mechanism is even more restrictive, and we could not tune it to work in this environment. Overall, with the efficient Sinkhorn Step facilitating individual waypoint exploration, MPOT outperforms the baselines in terms of run time complexity by a considerable margin in this case while maintaining reasonable smoothness and task performance.
>
> - **How is $D^P$ initialized at the beginning of the algorithm?** In this paper, we adopt the common regular polytope (i.e., simplex, orthoplex, and hypercube) known to exist for any dimension. Their vertex coordinate computations are well-known in Geometry literature. Hence, for completeness, we provide the construction description of polytope vertices in Appendix F. For actual implementation, we also provide the example code in the supplementary material. The vertex construction functions are found in `mpot/mpot/utils/polytopes.py`. Interestingly, we conducted an ablation study and showed that polytope structure and random rotation are essential for sample efficiency in terms of search directions compared to just sampling random search directions on an $d$-dimensional sphere, as reflected in Table 5 in Appendix J3.

---

> > ### Comment · Reviewer_MARJ · 2023-08-19
> > **Good job**
> >
> > Thank you for fully addressing my questions. 36D is high-dimensional, and the new result is definitely impressive. I’ll raise the score to 7.

---

> ### Comment · Area_Chair_ycqk · 2023-08-19
>
> Please be sure to read the authors' response to your initial review and reply indicating the extent to which it resolves your initial questions and concerns.

---

### Official Review · Reviewer_caUS · 2023-07-07

**Soundness:** 3 good
**Presentation:** 3 good
**Contribution:** 3 good
**Rating:** 7
**Confidence:** 2

**Summary:**

The paper proposes a trajectory optimization method using Sinkhorn Step which is able to perform efficient gradient-free batch optimization with non-linear objectives.

**Strengths:**

Gradient-free motion optimization is an important area with broad potential applications.

The contribution of the paper is novel. An original update rule is proposed, which significantly improves the converging speed and trajectory quality.

The paper is well-structured. The experiment results are adequate and easy to understand.

**Weaknesses:**

It is unclear how the proposed method is compared to learning-based trajectory planning approaches, and also how large is the improvement of the proposed method compared to existing optimization methods when using learning-based trajectory predictions as initializations.

**Questions:**

See weaknesses.

**Limitations:**

Limitations are well-discussed in the paper.

---

> ### Author Rebuttal · Authors · 2023-08-08
>
> We thank the reviewer for appreciating our paper contributions.
>
> Learning-based motion planning methods [1] typically utilize a dataset from previously successfully generated plans to learn generative priors for generating plans directly [2] or using rollout samples as initialization for motion planners [3]. Differently from the learning-based setting, we propose an optimization-based planner without any learning elements, and we conduct comparison experiments with other widely-used motion planners to support the paper's claims.
>
> Per the reviewer's suggestion, a learning-based setting can naturally complement MPOT to provide even better initializations, as currently, we only use GP priors to provide random initial smooth trajectories. However, this is considered for future work. In the current research, we focus on the proposition of the novel operator, the Sinkhorn Step, that allows us to tackle motion planning problems, resulting in an efficient trajectory optimization method producing smooth trajectories. This opens up exciting future directions for learning to plan applications, e.g., using MPOT as an expert for learning to plan or using implicit or explicit learned policies to improve MPOT further.
>
> [1] J. Wang et al., "A survey of learning-based robot motion planning," IET Cyber-Systems and Robotics, vol. 3, no. 4, pp. 302–314, 2021.
>
> [2] A. H. Qureshi, A. Simeonov, M. J. Bency, and M. C. Yip, "Motion planning networks," in IEEE ICRA, 2019.
>
> [3] J. Urain, A. Le, A. Lambert, G. Chalvatzaki, B. Boots, and J. Peters, "Learning implicit priors for motion optimization," in IEEE/RSJ International Conference on Intelligent Robots and Systems, 2022.

---

### Author Rebuttal · Authors · 2023-08-08

We thank the reviewers for their time and effort in reviewing our paper, and their constructive and positive evaluations of our work. Below are the main questions and concerns from the reviewers that we have addressed. We briefly state the discussions as follows (full answers can be found in the reviewers' rebuttal boxes):

- **Comparison with learning-based motion planning methods**

     Learning-based motion planning methods typically utilize a dataset from previously successfully generated plans to learn generative priors for generating plans directly or using rollout samples as initialization for motion planners. In fact, the learning-based methods can naturally complement MPOT to provide even better initializations, as currently, we only use GP priors to provide random initial smooth trajectories. However, this is considered for future work. In the current scope, we focus on the proposition of the novel operator, the Sinkhorn Step, that allows us to tackle motion planning problems, resulting in an efficient trajectory optimization method producing smooth trajectories.


- **On the dimensionality of trajectory optimization problem**

    We do not consider any movement primitives in our experiments. In all cases, we consider optimizing the full-state (the concatenation of position and velocity) trajectories batch-wise. As mentioned in the experiment description (Sec. 5.1), the state dimension (configuration position and velocity) is $d=4$ for the point-mass experiment, $d=14$ for the Panda experiment, and $d=36$ ($3$ dimensions for the base, $1$ for the torso, and $14$ for the two arms, and their velocities) for the mobile manipulation experiment. Note that the typical optimizing variable would be the whole trajectory, leading to variable dimension $T \times d$. In our vectorization setting, it would be batched $b \times T \times d$. Hence, even for the Panda case, typical full-state motion planning with multiple objectives, such as obstacle avoidance, self-collision avoidance, smoothness, goal-reaching, joint limit handling, etc., is inherently challenging due to many local minima and high computational costs for many objectives.


- **Baseline implementation considerations**

    We further describe our experiment settings in Appendix I. Striving for a fair comparison, in all cases, besides RRT*/I-RRT*, we reimplement all baselines in PyTorch and fine-tuned them with the vectorization option. To the best of our knowledge, the baselines are not explicitly designed for plan vectorization. Thus, their associated public codebases are implemented for single-plan querying.


- **Additional baseline comparisons in the TIAGo++ environment**

    We conducted additional comparative experiments on the TIAGo++ environment to better support our claims, and according to the reviewer's request, the results are available in the rebuttal attachment. Following the reviewers' suggestion, we tried SGPMP and observed that we could not tune SGPMP to surpass the success rate of 5/20; hence we opted for the minimum iterations to achieve the mentioned success rate. The problem lies in how SGPMP explores with constant GP variance in a high dimensional setting; thus, a possible future extension of SGPMP with adaptive explorative variance could mitigate the problem. Similarly, STOMP's exploration mechanism is even more restrictive, and we could not tune it to work in this environment. Overall, with the efficient Sinkhorn Step facilitating individual waypoint exploration, MPOT outperforms the baselines in terms of run time complexity by a considerable margin in this case while maintaining reasonable smoothness and task performance.

In summary, we propose Sinkhorn Step - a batch gradient-free optimization operator formulated as an Optimal Transport problem, leveraging the efficient Sinkhorn algorithm. We then apply the Sinkhorn Step to trajectory optimization, resulting in Motion Planning via Optimal Transport (MPOT) method. MPOT is inherently a multi-modality motion planner that optimizes a batch of high-dimensional smooth trajectories on multiple non-convex objectives, exhibiting individual waypoint exploration for better local minima escape. Finally, we also preliminarily investigate the Sinkhorn Step theoretical properties under standard assumptions.

---

### Decision · Program_Chairs · 2023-09-21

**Decision:**

Accept (poster)

**Comment:**

The paper proposes an optimal transport-based approach (MPOT) to motion planning capable of producing smooth trajectories without the need for objective gradients as are required by trajectory optimization-based methods. Integral to MPOT is the proposed zero-order parallelizable update rule that enables gradient-free optimization over a batch of trajectories. Smoothness is achieved using a Gaussian Process prior over the costs. Experiments on point-mass navigation and mobile manipulation domains demonstrate that MPOT outperforms contemporary baselines in terms of planning time and success rate.

The paper was well received by all four reviewers, who agree that real-time, gradient-free motion planning/optimization is an important problem. The proposed MPOT algorithm, including the Sinkhorn Step that enables batch gradient-free optimization, is sound and provides a novel contribution, and its effectiveness is empirically demonstrated. The reviewers initially raised a few questions/concerns including the potential limitations of requiring fixed-length sequences for batch optimization, which the authors largely resolved in their detailed responses.